# Exploring and Exploiting Decision Boundary Dynamics for Adversarial Robustness

**Yuancheng Xu**[†]  **Yanchao Sun**[†]  **Micah Goldblum**[‡]  **Tom Goldstein**[†]  **Furong Huang**[†]
[†] University of Maryland, College Park  [‡] New York University
[†]{ycxu,ycs,tomg,furongh}@umd.edu  [‡]goldblum@nyu.edu

## ABSTRACT

The robustness of a deep classifier can be characterized by its *margins*: the decision boundary's distances to natural data points. However, it is unclear whether existing robust training methods effectively increase the margin for each vulnerable point during training. To understand this, we propose a continuous-time framework for quantifying the relative speed of the decision boundary with respect to each individual point. Through visualizing the moving speed of the decision boundary under Adversarial Training, one of the most effective robust training algorithms, a surprising moving-behavior is revealed: the decision boundary moves away from some vulnerable points but simultaneously moves closer to others, decreasing their margins. To alleviate these *conflicting dynamics* of the decision boundary, we propose *Dynamics-Aware Robust Training* (DyART), which encourages the decision boundary to engage in movement that prioritizes increasing smaller margins. In contrast to prior works, DyART directly operates on the margins rather than their indirect approximations, allowing for more targeted and effective robustness improvement. Experiments on the CIFAR-10 and Tiny-ImageNet datasets verify that DyART alleviates the conflicting dynamics of the decision boundary and obtains improved robustness under various perturbation sizes compared to the state-of-the-art defenses. Our code is available at https://github.com/Yuancheng-Xu/Dynamics-Aware-Robust-Training.

## 1 INTRODUCTION

Deep neural networks have exhibited impressive performance in a wide range of applications (Krizhevsky et al., 2012; Goodfellow et al., 2014; He et al., 2016a). However, they have also been shown to be susceptible to adversarial examples, leading to issues in security-critical applications such as autonomous driving and medicine (Szegedy et al., 2013; Nguyen et al., 2015). To alleviate this problem, adversarial training (AT) (Madry et al., 2017; Shafahi et al., 2019; Zhang et al., 2019; Gowal et al., 2020) was proposed and is one of the most prevalent methods against adversarial attacks. Specifically, AT aims to find the worst-case adversarial examples based on some surrogate loss and adds them to the training dataset in order to improve robustness.

Despite the success of AT, it has been shown that over-parameterized neural networks still have insufficient model capacity for fitting adversarial training data, partly because AT does not consider the vulnerability difference among data points (Zhang et al., 2021). The vulnerability of a data point can be measured by its margin: its distance to the decision boundary. As depicted in Figure 1a, some data points have smaller margins and are thus more vulnerable to attacks. Since AT does not directly operate on the margins and it uses a pre-defined perturbation bound for all data points regardless of their vulnerability difference, it is unclear whether the learning algorithm can effectively increase the margin for each vulnerable point. Geometrically, we would like to know if the decision boundary moves away from the data points, especially the vulnerable ones. As illustrated in Figure 1b, there can exist conflicting dynamics of the decision boundary: it moves away from some vulnerable points but simultaneously moves closer to other vulnerable ones during training. This motivates us to ask:

**Question 1** *Given a training algorithm, how can we analyze the dynamics of the decision boundary with respect to the data points?*

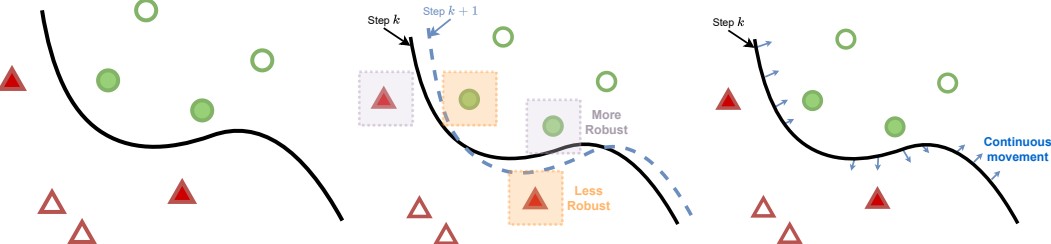

**(a)** The decision boundary, vulnerable (solid) and robust (hollow) points. **(b)** An update with conflicting impacts on robustness. **(c)** Continuous movement of the decision boundary.

**Figure 1:** The movement of the decision boundary. Red triangles and green circles are data points from two classes. Figure 1a shows the vulnerability difference among the data points: some are closer to the decision boundary, whereas others are farther from it. In Figure 1b, the decision boundary after an update moves away from some vulnerable points (made more robust) but simultaneously moves closer to other vulnerable ones (made less robust). Figure 1c describes the continuous movement of the decision boundary in Figure 1b.

To answer the above question, we propose a continuous-time framework that quantifies the instantaneous movement of the decision boundary as shown in Figure 1c. Specifically, we define the relative speed of the decision boundary w.r.t. a point to be the time derivative of its margin, which can be interpreted as the speed of its closest adversarial example moving away from it. We show that the speed can be derived from the training algorithm using a closed-form expression.

Using the proposed framework, we empirically compute the speed of the decision boundary w.r.t. data points for AT. As will be shown in Figure 3, the aforementioned conflicting dynamics of the decision boundary (Figure 1b,1c) is revealed: the decision boundary moves *towards* many vulnerable points during training and decrease their margins, directly counteracting the objective of robust training. The desirable dynamics of the decision boundary, on the other hand, should increase the margins of all vulnerable points. This leads to another question:

**Question 2** *How to design algorithms that encourage the decision boundary to engage in movements that increase margins for vulnerable points, and not decrease them?*

To this end, we propose *Dynamics-Aware Robust Training (DyART)*, which prioritizes moving the decision boundary away from more vulnerable points and increasing their margins. Specifically, DyART directly operates on margins of training data and carefully designs its cost function on margins for more desirable dynamics. Note that directly optimizing margins in the input space is technically challenging since it was previously unclear how to compute the gradient of the margin. In this work, we derive the closed-form expression for the gradient of the margin and present an efficient algorithm to compute it, making gradient descent viable for DyART. In addition, since DyART directly operates on margins instead of using a pre-defined uniform perturbation bound for training as in AT, DyART is naturally robust for a wide range of perturbation sizes $\epsilon$. Experimentally, we demonstrate that DyART mitigates the conflicting dynamics of the decision boundary and achieves improved robustness performance on diverse attacking budgets.

**Summary of contributions.** **(1)** We propose a continuous-time framework to study the relative speed of the decision boundary w.r.t. each individual data point and provide a closed-form expression for the speed. **(2)** We visualize the speed of the decision boundary for AT and identify the conflicting dynamics of the decision boundary. **(3)** We present a close-form expression for the gradient of the margin, allowing for direct manipulation of the margin. **(4)** We introduce an efficient alternative to compute the margin gradient by replacing the margin with our proposed *soft margin*, a lower bound of the margin whose approximation gap is controllable. **(5)** We propose Dynamics-Aware Robust Training (DyART), which alleviates the conflicting dynamics by carefully designing a cost function on soft margins to prioritize increasing smaller margins. Experiments show that DyART obtains improved robustness over state-of-the-art defenses on various perturbation sizes.

## 2 RELATED WORK

**Decision boundary analysis.** Prior works on decision boundary of deep classifiers have studied the small margins in adversarial directions (Karimi et al., 2019), the topology of classification regions

(Fawzi et al., 2018), the relationship between dataset features and margins (Ortiz-Jimenez et al., 2020) and improved robust training by decreasing the unwarranted increase in the margin along adversarial directions (Rade & Moosavi-Dezfooli, 2022). While these works study the static decision boundary of trained models, our work focuses on the decision boundary dynamics during training.

**Weighted adversarial training.**    Adversarial training and its variants (Madry et al., 2017; Zhang et al., 2019; Wang et al., 2019; Zhang et al., 2020b) have been proposed to alleviate the adversarial vulnerability of deep learning models. To better utilize the model capacity, weighted adversarial training methods are proposed (Zeng et al., 2020; Liu et al., 2021; Zhang et al., 2021) aiming to assign larger weights to more vulnerable points closer to the decision boundary. However, these methods rely on indirect approximations of margins that are not optimal. For example, GAIRAT (Zhang et al., 2021) uses the least number of iterations needed to flip the label of an clean example as an surrogate to its margin, which is shown to be likely to make wrong judgement on the robustness (Liu et al., 2021). As another approximation, the logit margin (Liu et al., 2021; Zeng et al., 2020) is used but larger logit margin values do not necessarily correspond to larger margins. In contrast, our proposed `DyART` directly uses margins to characterize the vulnerability of data points.

**Margin maximization.**    Increasing the distance between the decision boundary and data points has been discussed in prior works. The authors of Elsayed et al. (2018) propose to maximize the first order Taylor's expansion approximation of the margin at the clean data point, which is inaccurate and computationally prohibitive since it requires computing the Hessian of the classifier. The authors of Atzmon et al. (2019) propose to maximize the distance between each data point and some point on the decision boundary, which is not the closest one and thus does not increase the margin directly. MMA (Ding et al., 2020) uses the uniform average of cross-entropy loss on the closest adversarial examples as the objective function, indirectly increasing the average margins. All of these methods maximize the average margin indirectly and do not consider the vulnerability differences among points. In contrast, our proposed `DyART` will utilize our derived closed-form expression for margin gradient to directly operate on margins and moreover, prioritize increasing smaller margins.

## 3    PRELIMINARIES AND NOTATIONS

**Data and model.**    We consider a classification task with inputs $x \in \mathcal{X}$ and corresponding labels $y \in \mathcal{Y} = \{1, 2, ..., K\}$. A deep classifier parameterized by $\theta$ is denoted by $f_\theta(x) = \arg\max_{c \in \mathcal{Y}} z_\theta^c(x)$ where $z_\theta^c(x)$ is the logit for class $c$.

**Decision boundary.**    Denote the logit margin for class $y$ as follows:

$$\phi_\theta^y(x) = z_\theta^y(x) - \max_{y' \neq y} z_\theta^{y'}(x) \tag{1}$$

In this paper, we will use $\phi_\theta^y(x)$ and $\phi^y(x, \theta)$ interchangeably. Observe that $x$ is classified as $y$ if and only if $\phi_\theta^y(x) > 0$. Therefore, the zero level set of $\phi_\theta^y(x)$, denoted by $\Gamma_y = \{x : \phi_\theta^y(x) = 0\}$, is the decision boundary for class $y$. For a correctly classified point $(x, y)$, its margin $R_\theta(x)$ is defined to be the distance from $x$ to the decision boundary for class $y$. That is,

$$R_\theta(x) = \min_{\hat{x}} \|\hat{x} - x\|_p \quad \text{s.t.} \quad \phi_\theta^y(\hat{x}) = 0 \tag{2}$$

where $\| \cdot \|_p$ is the $\ell_p$ norm with $1 \leq p \leq \infty$.

**Difference between logit margin and margin.**    The logit margin $\phi_\theta^y(x)$ is the gap between the logits values that are *output* by the neural network. On the other hand, the margin $R_\theta(x)$ is the distance from the data point to the decision boundary, and is measured in the *input* space $\mathcal{X}$.

**Continuous-time formulation.**    To study the instantaneous movement of the decision boundary in Section 4, we will use the continuous-time formulation for the optimization on the parameters $\theta$, denoted as $\theta(t)$. Let $\theta'(t)$ be the continuous-time description of the update rule of the model parameters. When using gradient descent on a loss function $L$, we have $\theta'(t) = -\nabla_\theta L(\theta(t))$.

## 4    DYNAMICS OF THE DECISION BOUNDARY

In this section, we will study the dynamics of the decision boundary in continuous time. We first define its speed w.r.t. each data point, and then provide a closed-form expression for it. Finally, we visualize the speed of the decision boundary under Adversarial Training and analyze its dynamics.

### 4.1 SPEED OF THE DECISION BOUNDARY

Consider a correctly classified clean example $(x_i, y_i)$. Our goal is to capture the movement of the decision boundary $\Gamma_{y_i}(t) = \{x : \phi^{y_i}(x, \theta(t)) = 0\}$ w.r.t. $x_i$ as $t$ varies continuously. To this end, we consider the curve of the closest boundary point $\hat{x}_i(t)$ on $\Gamma_{y_i}(t)$ to $x_i$:

**Definition 1** (Curve of the closest boundary point $\hat{x}_i(\cdot)$). *Suppose that $(x_i, y_i)$ is correctly classified by $f_{\theta(t)}$ in some time interval $I$. Define the curve of the closest boundary point $\hat{x}_i(\cdot) : I \to \mathcal{X}$ as*

$$\hat{x}_i(t) = \arg\min_{\hat{x}} \|\hat{x} - x_i\|_p \quad s.t. \quad \phi^{y_i}(\hat{x}, \theta(t)) = 0. \tag{3}$$

*Define the margin of $x_i$ at time $t$ to be $R(x_i, t) = \|\hat{x}_i(t) - x_i\|_p$.*

An example of the curve of the closest boundary point is depicted in Figure 2. To understand how the distance between the decision boundary $\Gamma_{y_i}(t)$ and $x_i$ changes, it suffices to focus on the curve of the closest boundary point $\hat{x}_i(t)$. We define the speed of the decision boundary to be the time derivative of the margin as follows:

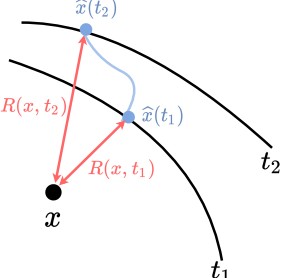

**Definition 2** (Speed of the decision boundary $s(x_i, t)$). *Under the setting of definition 1, define the speed of the decision boundary w.r.t. $x_i$ as $s(x_i, t) = \frac{d}{dt}R(x_i, t) = \frac{d}{dt}\|\hat{x}_i(t) - x_i\|_p$.*

Note that the speed $s(x_i, t) > 0$ means that the robustness is improving for $x_i$ at time $t$, which is desirable during robust training. The following proposition gives a closed-form expression for the speed, given a training algorithm $\theta'(t)$.

**Figure 2:** The curve of the closest boundary point $\hat{x}(t)$ (in blue) of the data point $x$.

**Proposition 3** (Closed-form expression of the speed $s(x_i, t)$). *Let $\hat{x}_i(t)$ be the curve of the closest boundary point w.r.t. $x_i$. For $1 \le p \le \infty$, the speed of decision boundary w.r.t. $x_i$ under $\ell_p$ norm is*

$$s(x_i, t) = \frac{1}{\|\nabla_x \phi^{y_i}(\hat{x}_i(t), \theta(t))\|_q} \nabla_\theta \phi^{y_i}(\hat{x}_i(t), \theta(t)) \cdot \theta'(t) \tag{4}$$

*where $q$ satisfies that $1/q + 1/p = 1$. In particular, $q = 1$ when $p = \infty$.*

**Remark.** Note that equation 4 is still valid when $\hat{x}_i(t)$ is just a locally closest boundary point (i.e. a local optimum of equation 3). In this case, $s(x_i, t)$ is interpreted as the moving speed of decision boundary around the locally closest boundary point $\hat{x}_i(t)$. The main assumption for equation 4 is the local differentiability of $\phi^{y_i}(\cdot, \theta(t))$ at $\hat{x}_i(t)$. The full assumptions, proof and discussions are left to Appendix B. Special care has been taken for $p = \infty$ since $\ell_\infty$ norm is not differentiable.

According to equation 4, the speed $s(x_i, t_0)$ is positive at time $t_0$ when $\nabla_\theta \phi^{y_i}(\hat{x}_i(t_0), \theta(t_0)) \cdot \theta'(t_0) > 0$, i.e., $\phi^{y_i}(\hat{x}_i(t), \theta(t))$ increases at time $t_0$, meaning that the boundary point $\hat{x}_i(t_0)$ will be correctly classified after the update. Also, the magnitude of the speed tends to be larger if $\|\nabla_x \phi^{y_i}(\hat{x}_i(t), \theta(t))\|_q$ is smaller, i.e., the margin function $\phi^{y_i}$ is flatter around $\hat{x}_i(t)$. In the remaining part of the paper, we will denote $s(x_i, t)$ and $R(x_i, t)$ as $s(x_i)$ and $R(x_i)$ when the indication is clear.

**Computing the closest boundary point.** We use the Fast Adaptive Boundary Attack (FAB) (Croce & Hein, 2020a) to compute $\hat{x}_i(t)$ in equation 4. Specifically, FAB iteratively projects onto the linearly approximated decision boundary with a bias towards the original data point, so that the resulting boundary point is close to the original point $x_i$. Note that FAB only serves as an algorithm to find $\hat{x}_i(t)$, and can be decoupled from the remaining part of the framework. In our experiments we find that FAB can reliably find locally closest boundary points given enough iterations, where the speed expression in equation 4 is still valid. We give more details of how to check the local optimality condition of equation 3 and the performance of FAB in Appendix C.1. Note that in Section 5.2, we will see that directly using FAB is computationally prohibitive for robust training and we will propose a more efficient solution. In the next section, we will still use FAB to find closest boundary points for more accurate visualization of the dynamics during adversarial training.

### 4.2 DYNAMICS OF ADVERSARIAL TRAINING

In this section, we numerically investigate the dynamics of the decision boundary during adversarial training. We visualize the speed and identify the conflicting dynamics of the decision boundary.

**Experiment setting.** To study the dynamics of AT in different stages of training where models have different levels of robustness, we train a ResNet-18 (He et al., 2016a) with group normalization (GN) (Wu & He, 2018) on CIFAR-10 using 10-step PGD under $\ell_\infty$ perturbation with $\epsilon = \frac{8}{255}$ from two pretrained models: (1) a partially trained model using natural training with $85\%$ clean accuracy and $0\%$ robust accuracy; (2) a partially trained model using AT with $75\%$ clean accuracy and $42\%$ robust accuracy under 20-step PGD attack. Note that we replace the batch normalization (BN) layers with GN layers since the decision boundaries are not the same during training and evaluation when BN is used, which can cause confusion when studying the dynamics of the decision boundary. On both pretrained models, we run one iteration of AT on a batch of training data. For correctly classified points in the batch of data, we compute the margins as well as the speed of the decision boundary.

**Conflicting dynamics.** The dynamics of the decision boundary on both pretrained models under AT is shown in Figure 3. The speed values are normalized so that the maximum absolute value is 1 for better visualization of their relative magnitude. We can observe that on both pretrained models, the decision boundary has negative speed w.r.t. a significant proportion of non-robust points with $R(x_i) < \frac{8}{255}$. That is, the margins of many vulnerable points *decrease* during adversarial training even though the current update of the model is computed on these points, which counteracts the objective of robust training. In the next section, we will design a dynamics-aware robust training method to mitigate such conflicting dynamics issue.

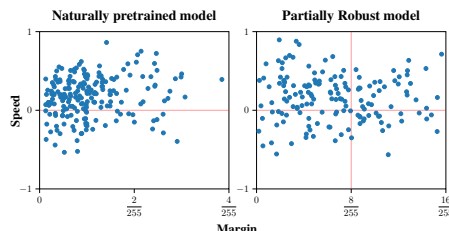

**Figure 3:** Margin-speed plot of AT on a training batch. Among points with margins smaller than $\frac{8}{255}$, there are $28.8\%$ and $29.4\%$ points with negative speed on each pretrained model.

## 5 DyART: Dynamics-aware Robust Training

In this section, we propose Dynamics-Aware Robust Training (DyART) to mitigate the conflicting dynamics issue. In Section 5.1, we show how to design an objective function to prioritize improving smaller margins and how to compute the gradient of such objective. In Section 5.2, we overcome the expensive cost of finding the closest boundary points and present the full DyART algorithm.

### 5.1 Objective for desirable dynamics

We aim to design a loss function $L^R(\theta)$ to directly increase the overall margins for effective robustness improvement. We propose to use the robustness loss $L^R(\theta) := \mathbb{E}_x[h(R_\theta(x))]$, where $h : \mathbb{R} \to \mathbb{R}$ is a carefully selected *cost function* that assigns a cost value $h(R)$ to a margin $R$. When designing $h(\cdot)$, it is crucial that minimizing $L^R(\theta) = \mathbb{E}_x[h(R_\theta(x))]$ encourages the *desirable dynamics* of the decision boundary: the decision boundary has positive speed w.r.t. vulnerable points with small margins.

**Dynamics-aware loss function.** To design such a *dynamics-aware* loss function, the following two properties of the cost function $h$ are desired. **(1) Decreasing** (i.e., $h'(\cdot) < 0$): a point with a smaller margin should be assigned a higher cost value since it is more vulnerable. **(2) Convex** (i.e., $h''(\cdot) > 0$): the convexity condition helps prioritize improving smaller margins. To see this, consider minimizing the loss function $L^R(\theta)$ on $m$ points $\{x_i, y_i\}_{i=1}^m$ with margins $\{R_i\}_{i=1}^m$, where the objective becomes $\frac{1}{m}\sum_{i=1}^m h(R_\theta(x_i))$. At each iteration, the optimizer should update the model to decrease the objective value. Therefore, in continuous-time we have that $\frac{d}{dt}\sum_{i=1}^m h(R(x_i, t)) < 0$. Using the chain rule and the definition that the speed $s(x_i, t) = \frac{d}{dt}R(x_i, t)$, we obtain that $\sum_{i=1}^m h'(R_i)s(x_i, t) < 0$. Given that $h'(\cdot) < 0$, the ideal case is that $s(x_i, t) > 0$ for all $x_i$ and thus the sum $\sum_{i=1}^m h'(R_i)s(x_i, t) < 0$. In this case, the margins of all data points increase. However, due to the existence of conflicting dynamics as described in Section 4.2, some points may have negative speed $s(x_i, t) < 0$ while $\sum_{i=1}^m h'(R_i)s(x_i, t)$ stays negative. In the presence of such conflicting dynamics, if $|h'(R_i)|$ is large (i.e., $h'(R_i)$ is small since $h'(\cdot) < 0$), it is more likely that $s(x_i, t) > 0$ since otherwise it is harder to make $\sum_{i=1}^m h'(R_i)s(x_i, t)$ negative. When $h''(\cdot) > 0$, a smaller margin $R_i$ has smaller $h'(R_i)$ and thus $s(x_i, t)$ tends to be positive. Therefore, requiring $h''(\cdot) > 0$ incentivizes the decision boundary to have positive speed w.r.t. points with smaller margins.

How to design the optimal $h(\cdot)$ is still an open problem. In this paper, we propose to use

$$h(R) = \begin{cases} \frac{1}{\alpha}\exp(-\alpha R), & R < r_0 \\ 0, & \text{otherwise} \end{cases} \tag{5}$$

where the hyperparameters $\alpha > 0$ and $r_0 > 0$. Larger $\alpha$ values prioritize improving smaller margins. The threshold $r_0$ is used to avoid training on points that are too far away from the clean data points.

**Difficulties of computing margin gradient.** Directly minimizing $\mathbb{E}_x[h(R_\theta(x))]$ through gradient-based optimization methods requires computing the gradient $\nabla_\theta h(R_\theta(x_i))$ w.r.t. the model parameters. However, it was previously unclear how to compute $\nabla_\theta h(R_\theta(x_i))$, which partly explains why previous works did not directly operate on the margins. The difficulty of computing $\nabla_\theta h(R_\theta(x_i))$ lies in the fact that $R_\theta(x_i)$, as defined in equation 2, involves a constrained optimization problem and thus its gradient $\nabla_\theta R_\theta(x_i)$ cannot be computed straightforwardly. An additional challenge is dealing with the non-smoothness of the $\ell_\infty$ norm, which is widely used in the robust training literature.

**Our solution.** We overcome the above challenges and provide the following close-form expression for the gradient of any smooth function of the margin. The proof is provided in Appendix B.

**Theorem 4** (The gradient $\nabla_\theta h(R_\theta(x_i))$ of any smooth function of the margin). *For $1 \le p \le \infty$,*

$$\nabla_\theta h(R_\theta(x_i)) = \frac{h'(R_\theta(x_i))}{\|\nabla_x \phi^{y_i}(\hat{x}_i, \theta)\|_q} \nabla_\theta \phi^{y_i}(\hat{x}_i, \theta) \tag{6}$$

*where $q$ satisfies that $1/q + 1/p = 1$. In particular, $q = 1$ when $p = \infty$.*

Note that another expression for the margin gradient (i.e., $h$ is the identity function in equation 6) was derived in MMA (Ding et al., 2020), with the following distinctions from us: (a) The expression in MMA does not apply to the $\ell_\infty$ norm while ours does. (b) The coefficient $\frac{1}{\|\nabla_x \phi^{y_i}(\hat{x}_i, \theta)\|_q}$ in our expression is more informative and simpler to compute. (c) MMA treats the aforementioned coefficient as a constant during training, and therefore does not properly follow the margin gradient.

Computing $\nabla_\theta h(R_\theta(x_i))$ requires computing the closest boundary points $\hat{x}_i$, which can be computationally prohibitive for robust training. In the next section, we propose to use the closest point $\hat{x}_i^{\text{soft}}$ on the *soft* decision boundary instead, whose quality of approximation to the exact decision boundary is controllable and computational cost is tractable. We will then present the full DyART algorithm.

## 5.2 EFFICIENT ROBUST TRAINING

**Directly finding the closest boundary points is expensive.** Since the closest boundary point $\hat{x}_i$ can be on the decision boundary between the true class and any other class, FAB needs to form a linear approximation of the decision boundary between the true class and every other class at each iteration. This requires computing the Jacobian of the classifier, and the computational cost scales linearly with the number of classes $K$ (Croce & Hein, 2020b). Therefore, finding the closest points on the exact decision boundary is computationally prohibitive for robust training in multi-class classification settings, especially when $K$ is large. To remedy this, we propose to instead use the closest points on the *soft decision boundary* as elaborated below.

**Soft decision boundary.** We replace the maximum operator in logit margin (equation 1) with a smoothed maximum controlled by the temperature $\beta > 0$. Specifically, we define the soft logit margin of the class $y$ as

$$\Phi_\theta^y(x; \beta) = z_\theta^y(x) - \frac{1}{\beta} \log \sum_{y' \ne y} \exp(\beta z_\theta^{y'}(x)) \tag{7}$$

The soft decision boundary is defined as the zero level set of the soft logit margin: $\Gamma_y^{\text{soft}} = \{x : \Phi_\theta^y(x; \beta) = 0\}$. For $x_i$ with $\Phi_\theta^{y_i}(x_i; \beta) > 0$, the closest soft boundary point is defined as

$$\hat{x}_i^{\text{soft}} = \arg\min_{\hat{x}} \|\hat{x} - x_i\|_p \quad \text{s.t.} \quad \Phi_\theta^y(\hat{x}; \beta) = 0, \tag{8}$$

and the *soft margin* is defined as $R_\theta^{\text{soft}}(x_i) = \|\hat{x}_i^{\text{soft}} - x_i\|_p$. Note that we do not define $R^{\text{soft}}(x_i)$ when $\Phi_\theta^{y_i}(x_i; \beta) < 0$. The relationship between the exact and soft decision boundary is characterized by the following proposition:

**Proposition 5.** *If $x$ is on the soft decision boundary $\Gamma_y^{\text{soft}}$, i.e. $\Phi_\theta^y(x; \beta) = 0$, then $\frac{\log(K-1)}{\beta} \ge \phi_\theta^y(x) \ge 0$. Moreover, when $\Phi_\theta^{y_i}(x_i; \beta) > 0$, then $R_\theta^{\text{soft}}(x_i) \le R_\theta(x_i)$.*

In other words, the soft decision boundary is always closer to $x_i$ than the exact decision boundary as shown in Figure 4. Moreover, the quality of approximation to the exact decision boundary is controllable: the gap between the two decreases as $\beta$ increases and vanishes when $\beta \to \infty$. Therefore, increasing the soft margins will increase the exact margins as well.

**Benefits of the soft decision boundary.** *(1) Computational efficiency.* Using the smoothed max operator, finding the closest soft boundary point does not require forming linear approximations for the decision boundary between the true class and all the other classes anymore. We adapt the FAB method to solve for $\hat{x}_i^{\text{soft}}$ (see details in Appendix C.2). Its computational cost for each iteration is independent of the number of classes $K$, which is the same as the PGD training. *(2) Effective information usage.* Another benefit of using the smoothed max operator in soft logit margin is that, unlike the logit margin $\phi_\theta^{y_i}(x_i)$, the soft logit margin $\Phi_\theta^{y_i}(x_i; \beta)$ contains information of logit values of all classes. Therefore, the information of all classes is used at each iteration when finding $\hat{x}_i^{\text{soft}}$.

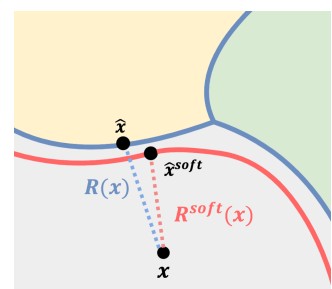

**Figure 4:** Exact decision boundary (in blue) for three classes (yellow, green and grey regions) and the soft decision boundary (in red) for the class of $x$.

**Loss function and its gradient.** The overall objective of DyART is to increase the soft margins and also achieve high clean accuracy. Denote a training data batch $\mathcal{B}$ of size $n$ and $\mathcal{B}_\theta^+$ of size $m$ to be $\{i \in \mathcal{B} : \Phi_\theta^{y_i}(x_i; \beta) > 0\}$. Our proposed method DyART uses the following loss function

$$L_\theta(\mathcal{B}) = \frac{1}{n} \sum_{i \in \mathcal{B}} l(x_i, y_i) + \frac{\lambda}{n} \sum_{i \in \mathcal{B}_\theta^+} h(R_\theta^{\text{soft}}(x_i))$$

where the first term is the average cross-entropy loss on natural data points and the second term is for increasing the soft margins. The hyperparameter $\lambda$ balances the trade-off between clean and robust accuracy. By applying equation 6, the gradient of the objective can be computed as

$$\nabla_\theta L_\theta(\mathcal{B}) = \frac{1}{n} \sum_{i \in \mathcal{B}} \nabla_\theta l(x_i, y_i) + \frac{\lambda}{n} \sum_{i \in \mathcal{B}_\theta^+} \frac{h'(R_\theta^{\text{soft}}(x_i))}{\|\nabla_x \Phi_\theta^{y_i}(\hat{x}_i^{\text{soft}}; \beta)\|_q} \nabla_\theta \Phi_\theta^{y_i}(\hat{x}_i^{\text{soft}}; \beta) \quad (9)$$

Since the soft margin $R_\theta^{\text{soft}}(x_i)$ is only defined for $x_i$ with $\Phi_\theta^{y_i}(x_i; \beta) > 0$, DyART requires training on a pretrained model with a relatively high proportion of points with positive $\Phi_\theta^{y_i}$ values. In practice, we find that a burn-in period of several epochs of natural training is enough for such pretrained model.

**Novelty compared with prior works. (1) Direct** and **efficient** manipulation of the margin. **(1a)** In contrast to prior works that depend on indirect approximations of margins, DyART directly operates on margins by utilizing the closed-from expression for the margin gradient in equation 6 whose computation was previously unclear. **(1b)** We significantly reduce the computational cost of computing margins and its gradients by introducing the *soft margin*, a lower bound of the margin whose approximation gap is controllable. **(2) Prioritizing** the growth of smaller margins by carefully designing the cost function $h(\cdot)$ to mitigate the conflicting dynamics. Therefore, DyART achieves more targeted and effective robustness improvement by directly and efficiently operating on margins as well as prioritizing the growth of smaller margins.

## 6 EXPERIMENTS

In this section, we empirically evaluate the effectiveness and performance of the proposed DyART on the CIFAR-10 (Krizhevsky et al., 2009) and Tiny-ImageNet (Deng et al., 2009) datasets. In Section 6.1, we evaluate the adversarial robustness of DyART and compare it with several state-of-the-art baselines. In Section 6.2, we visualize the dynamics of the decision boundary under DyART and analyze how it alleviates the conflicting dynamics.

### 6.1 ROBUSTNESS EVALUATION

**Architectures and training parameters.** In the experiments on the CIFAR-10 dataset, we use the Wide Residual Network (Zagoruyko & Komodakis, 2016) with depth 28 and width factor 10 (WRN-

28-10). On the Tiny-ImageNet dataset, we use pre-activation ResNet-18 (He et al., 2016b). Models are trained using stochastic gradient descent with momentum 0.9 and weight decay 0.0005 with batch size 256 for 200 epochs on CIFAR-10 and 100 epochs on Tiny-ImageNet. We use stochastic weight averaging (Izmailov et al., 2018) with a decay rate of 0.995 as in prior work (Gowal et al., 2020). We use a cosine learning rate schedule (Loshchilov & Hutter, 2016) without restarts where the initial learning rate is set to 0.1 for all baselines and DyART. To alleviate robust overfitting (Rice et al., 2020), we perform early stopping on a validation set of size 1024 using projected gradient descent (PGD) attacks with 20 steps.

**Baselines.** On CIFAR-10, the baselines include: (1) standard adversarial training (AT) (Madry et al., 2017) which trains on the worst case adversarial examples; (2) TRADES (Zhang et al., 2019) which trades off between the clean and robust accuracy; (3) MMA (Ding et al., 2020) which uses cross-entropy loss on the closest boundary points; (4) GAIRAT (Zhang et al., 2021) which reweights adversarial examples based on the least perturbation iterations. (5) MAIL (Liu et al., 2021) which reweights adversarial examples based on their logit margins. (6) AWP (Wu et al., 2020) which adversarially perturbs both inputs and model parameters. On Tiny-ImageNet, we compare with AT, TRADES, and MART whose hyperparameter settings are available for this dataset. The hyperparameters of the baselines and full experimental settings are found in Appendix D.1.

**Evaluation details.** We evaluate DyART and the baselines under $\ell_\infty$ norm constrained perturbations. The final robust accuracy is reported on AutoAttack (AA) (Croce & Hein, 2020b). For all methods, we choose the hyperparameters to achieve the best robust accuracy under the commonly used perturbation bound $\epsilon = \frac{8}{255}$. To fully compare the robustness performance among different methods, we report the robust accuracy under four additional perturbation bounds: $\frac{2}{255}, \frac{4}{255}, \frac{12}{255}$ and $\frac{16}{255}$.

**Hyperparameters of DyART.** We use the cost function $h(\cdot)$ in equation 5. On CIFAR-10, we use $\alpha = 3, r_0 = \frac{16}{255}, \lambda = 1000$ and apply gradient clipping with threshold 0.1. On Tiny-ImageNet, we use $\alpha = 5, r_0 = \frac{32}{255}, \lambda = 500$ and apply gradient clipping with threshold 1. The temperature $\beta$ is set to 5. We use 20 iterations to find the closest soft boundary points using the adapted version of FAB. We use 10 epochs of natural training as the burn-in period.

| Defense | Clean | $\epsilon = \frac{2}{255}$ | $\epsilon = \frac{4}{255}$ | $\epsilon = \frac{8}{255}$ | $\epsilon = \frac{12}{255}$ | $\epsilon = \frac{16}{255}$ |
|---|---|---|---|---|---|---|
| AT | $85.65 \pm 0.25$ | $79.08 \pm 0.12$ | $71.24 \pm 0.28$ | $53.20 \pm 0.16$ | $32.94 \pm 0.32$ | $16.12 \pm 0.23$ |
| TRADES | $82.92 \pm 0.30$ | $77.69 \pm 0.16$ | $70.68 \pm 0.15$ | $54.28 \pm 0.19$ | $\mathbf{36.65} \pm 0.24$ | $\mathbf{21.59} \pm 0.31$ |
| MART | $83.37 \pm 0.25$ | $76.58 \pm 0.24$ | $70.19 \pm 0.18$ | $52.91 \pm 0.24$ | $35.16 \pm 0.13$ | $18.80 \pm 0.14$ |
| MMA | $83.22 \pm 0.38$ | $74.24 \pm 0.52$ | $64.42 \pm 0.29$ | $44.02 \pm 0.33$ | $26.45 \pm 0.21$ | $13.78 \pm 0.25$ |
| GAIRAT | $\mathbf{86.59} \pm 0.31$ | $76.72 \pm 0.28$ | $64.64 \pm 0.25$ | $38.16 \pm 0.32$ | $19.01 \pm 0.18$ | $7.55 \pm 0.17$ |
| MAIL-TRADES | $83.96 \pm 0.52$ | $77.65 \pm 0.33$ | $69.11 \pm 0.35$ | $50.14 \pm 0.29$ | $31.57 \pm 0.24$ | $16.98 \pm 0.15$ |
| AWP | $84.27 \pm 0.19$ | $78.33 \pm 0.21$ | $70.82 \pm 0.26$ | $53.92 \pm 0.17$ | $35.24 \pm 0.26$ | $20.40 \pm 0.14$ |
| DyART | $85.55 \pm 0.24$ | $\mathbf{79.21} \pm 0.14$ | $\mathbf{71.73} \pm 0.18$ | $\mathbf{54.69} \pm 0.14$ | $35.74 \pm 0.25$ | $20.79 \pm 0.18$ |

**Table 1:** Clean and robust accuracy on CIFAR-10 under AA with different perturbation sizes on WRN-28-10.

| Defense | Clean | $\epsilon = \frac{2}{255}$ | $\epsilon = \frac{4}{255}$ | $\epsilon = \frac{8}{255}$ | $\epsilon = \frac{12}{255}$ | $\epsilon = \frac{16}{255}$ |
|---|---|---|---|---|---|---|
| AT | $48.09 \pm 0.38$ | $38.82 \pm 0.26$ | $30.18 \pm 0.27$ | $16.46 \pm 0.19$ | $7.74 \pm 0.20$ | $3.05 \pm 0.17$ |
| TRADES | $46.68 \pm 0.30$ | $37.84 \pm 0.21$ | $29.85 \pm 0.19$ | $16.76 \pm 0.17$ | $8.97 \pm 0.23$ | $4.43 \pm 0.11$ |
| MART | $45.51 \pm 0.29$ | $36.68 \pm 0.34$ | $29.15 \pm 0.25$ | $17.79 \pm 0.15$ | $9.91 \pm 0.17$ | $5.31 \pm 0.17$ |
| DyART | $\mathbf{49.71} \pm 0.18$ | $\mathbf{39.30} \pm 0.14$ | $\mathbf{30.69} \pm 0.21$ | $\mathbf{18.02} \pm 0.18$ | $\mathbf{10.08} \pm 0.09$ | $\mathbf{5.65} \pm 0.12$ |

**Table 2:** Clean and robust accuracy on Tiny-ImageNet under AA with different perturbation sizes on ResNet-18.

**Performance.** The evaluation results on CIFAR-10 and Tiny-ImageNet are shown in Table 1 and Table 2, respectively. On CIFAR-10, under three out of five perturbation bounds, DyART achieves the best robustness performance among all baselines. On Tiny-ImageNet, DyART obtains the highest robust accuracy under all perturbation bounds and also achieves the highest clean accuracy. These results indicate the superiority of DyART in increasing the margins. **(1)** Specifically, on CIFAR-10, DyART achieves the highest robust accuracy under $\epsilon = \frac{2}{255}, \frac{4}{255}$ and $\frac{8}{255}$, and achieves the second highest robust accuracy under $\epsilon = \frac{12}{255}$ and $\frac{16}{255}$, which is lower than TRADES. (1a) Since DyART prioritizes increasing smaller margins which are more important, DyART performs better than TRADES under smaller perturbation bounds and achieves much higher clean accuracy. (1b)

Although GAIRAT and AT have higher clean accuracy than `DyART`, their robustness performance is lower than `DyART` under all perturbation bounds. (1c) Thanks to directly operating on margins in the input space and encourage robustness improvement on points with smaller margins, `DyART` performs better than GAIRAT and MAIL-TRADES which use indirect approximations of the margins. **(2)** On Tiny-ImageNet, `DyART` achieves the best clean accuracy and the best robust accuracy under all perturbation bounds. Further experimental results on both datasets using various hyperparameter settings and types of normalization layers are left to Appendix D.2. We also provide results of `DyART` when trained with additional data from generated models (Rebuffi et al., 2021) in Appendix D.3.

## 6.2 DYNAMICS OF `DyART`

In this section we provide further insights into how `DyART` encourages the desirable dynamics by comparing it with adversarial training.

**Experimental setting.** To compare the dynamics of the decision boundary during training using `DyART` and AT, we empirically compute the margins and speed values for both methods. For fair comparison, we run `DyART` and AT on the same pretrained models for one iteration on the same batch of training data points. The pretrained models include a partially trained model using natural training and a partially trained model using AT, which are the same as in Section 4.2. For all the correctly classified points in this batch, we compute the margins and speed values under both methods. Note that the speed and margins correspond to the exact decision boundary, instead of the soft decision boundary used by `DyART` for robust training. Since both methods train on the same model and the same batch of data at this iteration, the margins are the same and only the speed values differ, which corresponds to the difference in dynamics of the decision boundary.

**`DyART` mitigates the conflicting dynamics.** We visualize the dynamics on both pretrained models under `DyART` and AT in Figure 5. Specifically, we divide the range of margins into multiple intervals and compute the proportion of positive and negative speed within all the correctly classified points. On the naturally pretrained model, most of the points have margins less than $\frac{4}{255}$ (the first bin) and are considered more vulnerable. Among these points, `DyART` reduces the proportion of the negative speed from $29.2\%$ to $15.3\%$ when comparing with AT. Therefore, a higher percentage of the margins of vulnerable

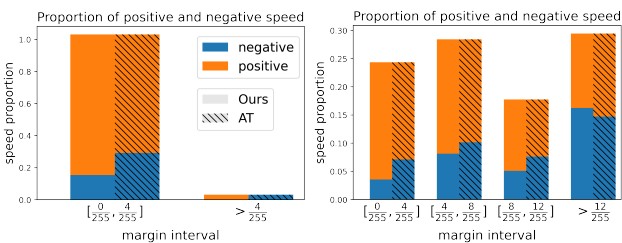

**(a)** Naturally pretrained model  **(b)** Partially robust model

**Figure 5:** Proportion of positive and negative speed values in each margin interval for AT and `DyART` on a naturally pretrained model and a partially robust model. Observe that `DyART` has lower proportion of negative speed for points with small margins ($< \frac{8}{255}$).

points will increase using `DyART`. On the adversarially pretrained model, `DyART` reduces the proportion of negative speed values in the first three margin intervals and therefore has better dynamics of the decision boundary. We conclude that compared with AT, `DyART` leads to better dynamics of the decision boundary where increasing smaller margins is prioritized.

## 7 CONCLUSIONS AND DISCUSSIONS

This paper takes one more step towards understanding adversarial training by proposing a framework for studying the dynamics of the decision boundary. The phenomenon of conflicting dynamics is revealed, where the movement of decision boundary causes the margins of many vulnerable points to decrease and harms their robustness. To alleviate the conflicting dynamics, we propose *Dynamics-Aware Robust Training (`DyART`)* which prioritizes moving the decision boundary away from more vulnerable points and increasing their margins. Experiments on CIFAR-10 and Tiny-ImageNet demonstrate that `DyART` achieves improved robustness under various perturbation bounds. Future work includes (a) theoretical understanding of the dynamics of adversarial training; (b) developing more efficient numerical methods to find the closest boundary points for robust training.

ACKNOWLEDGMENTS

The authors would like to thank Zhen Zhang, Chen Zhu and Wenxiao Wang for helpful discussions over the ideas. This work is supported by National Science Foundation NSF-IIS-FAI program, DOD-ONR-Office of Naval Research, DOD Air Force Office of Scientific Research, DOD-DARPA-Defense Advanced Research Projects Agency Guaranteeing AI Robustness against Deception (GARD), Adobe, Capital One and JP Morgan faculty fellowships.

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

# Supplementary Material

## A ADDITIONAL RELATED WORK

**Decision boundary analysis**   In this paper, we mathematically characterizes the dynamics of decision boundaries and provide methods to directly compute and control the dynamics. Prior to this work, there are also some interesting studies on the dynamics of margins, though from different perspectives. Rade & Moosavi-Dezfooli (2022) point out that adversarial training leads to a superfluous increase in the margin along the adversarial directions, which can be a reason behind the trade-off between accuracy and robustness. Ortiz-Jimenez et al. (2020) investigate the relationship between data features and decision boundaries, and reveal several properties of CNNs and adversarial training. Their results show that adversarial training exploits the sensitivity and invariance of models to improve the robustness. Tramèr et al. (2020) studies invariance-based adversarial examples and expose a fundamental trade-off between commonly used sesitivity-based adversarial examples and the invariance-based ones, where the behaviors of decision boundaries are identified.

**Other Approaches to Improve Adversarial Training.**   Recent works (Najafi et al., 2019; Rebuffi et al., 2021; Gowal et al., 2021; 2020) have shown that the robust accuracy of adversarial training can be improved significantly with additional data from unlabeled datasets, data augmentation techniques and generative models. These approaches enhance the robustness of models by augmenting the dataset, which is orthogonal to our proposed algorithm that focus on how to optimize the model with the original dataset. Wu et al. (2020) show that model robustness is related to the flatness of weight loss landscape, which is implicitly achieved by commonly used adversarial learning techniques. Based on this insight, the authors propose to explicitly regularize the flatness of the weight loss landscape, which can improve the robust accuracy of existing adversarial training methods. Cui et al. (2021) propose to use logits from a clean model to guide the learning of a robust model, which leads to both high natural accuracy and strong robustness.

We note that our method focuses on a different perspective of adversarial training, i.e., dynamics of decision boundary, and can be combined with these techniques to further improve the robust accuracy of the model. The investigation of such combination is out of the scope of this paper, and will be addressed in our future work.

**Certifiable Robustness.**   There is an important line of work studying guaranteed robustness of neural networks. For example, convex relaxation of neural networks (Gowal et al., 2019; Zhang et al., 2018; Wong & Kolter, 2018; Zhang et al., 2020a; Gowal et al., 2018) bounds the output of a network while the input data is perturbed within an $\ell_p$ norm ball. Randomized smoothing (Cohen et al., 2019) is another certifiable defense which adds Gaussian noise to the input during test time. Croce et al. (2019) propose a provably robust regularization for ReLU networks that maximizes the linear regions of the classifier and the distance to the decision boundary. Note that certifiable robust radius is a strict lower bound of the margin, which is the focus of our work.

# B  PROOF OF THE CLOSED-FORM EXPRESSION FOR THE SPEED IN EQUATION 4 AND THE MARGIN GRADIENT IN EQUATION 6

In this section, our goal is to prove the closed-form expression equation 4 as well as the margin gradient in equation 6 and provide further discussions. We first provide two preliminary lemmas and present the mathematical assumptions. Then we rigorously derive the closed-form expressions. Finally, we discuss more about the expression and its assumptions.

**Lemma 6.** *For $1 \leq p \leq \infty$ and let $q$ satisfies $1/q + 1/p = 1$. Let $a$ be any fixed vector. Then*

$$\left\| \nabla_x \|x - a\|_p \right\|_q = 1$$

*Proof.* Without loss of generality, assume $a$ is the zero vector. Write the $k$-th component of $x$ as $x_k$.

*Case 1: $1 \leq p < \infty$*

By calculation, $\frac{\partial \|x\|_p}{\partial x_k} = (\frac{|x_k|}{\|x\|_p})^{p-1} \cdot \text{sign}(x_k)$. Since $q = \frac{p}{p-1}$, we have that

$$\sum_k |\frac{\partial \|x\|_p}{\partial x_k}|^q = \sum_k |(\frac{|x_k|}{\|x\|_p})^{p-1} \cdot \text{sign}(x_k)|^{\frac{p}{p-1}}$$
$$= \sum_k \frac{|x_k|^p}{\|x\|_p^p}$$
$$= 1$$

Therefore, $\left\| \nabla_x \|x\|_p \right\|_q = (\sum_k |\frac{\partial \|x\|_p}{\partial x_k}|^q)^{1/q} = 1$.

*Case 2: $p = \infty$*

In this case, $\nabla_x \|x\|_\infty$ is a one-hot vector (with the one being the position of the element of $x$ with the largest absolute value). Therefore, $\|\nabla_x \|x\|_\infty\|_1 = 1$.

$\square$

The following lemma deals with the optimality condition for $p = \infty$. Special care needs to be taken since $L_\infty$ norm is not a differentiable function.

**Lemma 7.** *Let $\hat{x}$ be a local optimum of the constrained optimization problem:*
$$\hat{x} = \arg\min_z \|x - a\|_\infty \quad s.t. \quad \phi(x) = 0,$$
*where $a$ is any fixed vector with $\phi(a) > 0$. Assume that $\phi$ is differentiable at point $\hat{x}$. Denote the coordinates set $\mathcal{J} = \{j : |\hat{x}_j - a_j| = \|\hat{x} - a\|_\infty\}$. Denote the $k$-th component of $\nabla_x \phi(\hat{x})$ as $\nabla_x \phi(\hat{x})_k$. Then (a) for $j \in \mathcal{J}$, $\nabla_x \phi(\hat{x})_j$ and $\hat{x}_j - a_j$ have opposite signs; (b) for $k \notin \mathcal{J}$, $\nabla_x \phi(\hat{x})_k = 0$.*

**Remark.** *If $\phi(a) < 0$, then (a) for $j \in \mathcal{J}$, $\nabla_x \phi(\hat{x})_j$ and $\hat{x}_j - a_j$ have the same sign; (b) for $k \notin \mathcal{J}$, $\nabla_x \phi(\hat{x})_k = 0$.*

*Proof. (a)* Consider the perturbation $\hat{x}(\epsilon) = \hat{x} + (0, \cdots, \epsilon_{j_1}, \cdots, \epsilon_{j_m}, \cdots, 0)$ where $\mathcal{J} = \{j_1, \cdots, j_m\}$ and $\epsilon$ is a $m$ dimensional vector with $j$-th component $\epsilon_j$. Since $\phi(a) > 0$ and $\hat{x}$ is a local optimum, $\|x - a\|_\infty < \|\hat{x} - a\|_\infty$ imply $\phi(x) > 0$ if $x$ is sufficiently close to $\hat{x}$. Therefore, if every $\epsilon_{j_i}$ is chosen so that $|\hat{x}_{j_i} + \epsilon_{j_i} - a_{j_i}| < |\hat{x}_{j_i} - a_{j_i}|$ (that is, $\epsilon_{j_i}$ has different sign from $\hat{x}_{j_i} - a_{j_i}$) and $\|\epsilon\|$ sufficiently small, then $\|\hat{x}(\epsilon) - a\| < \|\hat{x} - a\|$ and thus $\phi(\hat{x}(\epsilon)) > 0$.

On the other hand, by Taylor expansion and the fact that $\phi(\hat{x}) = 0$, we have that

$$\phi(\hat{x}(\epsilon)) = \sum_{j \in \mathcal{J}} \nabla\phi(\hat{x})_j \epsilon_j + \mathcal{O}(\|\epsilon\|^2)$$

Therefore, $\sum_{j \in \mathcal{J}} \nabla \phi(\hat{x})_j \epsilon_j > 0$ for any such $\epsilon$. By taking other $\epsilon_k \to 0$ if necessary, we obtain that $\forall j \in \mathcal{J}, \nabla \phi(\hat{x})_j \epsilon_j \geq 0$, where $\epsilon_j$ has different sign from $\hat{x}_j - a_j$. Therefore, $\nabla \phi(\hat{x})_j$ and $\hat{x}_j - a_j$ have different signs.

*(b)* Take any $k \notin \mathcal{J}$ and consider the perturbation $\hat{x}(\epsilon) = \hat{x} + (0, \cdots, \epsilon_{j_1}, \cdots, \epsilon_k, \cdots, \epsilon_{j_k}, \cdots, 0)$ where $\epsilon = (\epsilon_{j_1}, \cdots, \epsilon_k, \cdots, \epsilon_{j_k})$. Choose any $\epsilon$ so that $\|\epsilon\|$ is sufficiently small, each $\epsilon_{j_i}$ has the opposite sign of $\hat{x}_{j_i} - a_{j_i}$ and $\epsilon_k$ small enough (which can be positive or negative), we have that $\phi(\hat{x}(\epsilon)) > 0$ since $\|\hat{x}(\epsilon) - a\|_\infty < \|\hat{x} - a\|_\infty$. By Taylor expansion, $\sum_{j \in \mathcal{J}} \nabla \phi(\hat{x})_j \epsilon_j + \epsilon_k \nabla_x \phi(\hat{x})_k > 0$ for any such $\epsilon$. By taking $\epsilon_j \to 0$ and using the fact that $\epsilon_k$ can be positive or negative, we conclude that $\nabla_x \phi(\hat{x})_k = 0$.

$\square$

Now we are ready to derive the closed-form expression of the speed. We first provide the full assumptions, then derive the expression, and finally we will discuss more about the assumptions. We will write $\hat{x}_i(t)$ as $\hat{x}_i$ when the indication is clear.

**Assumption 8.** *Suppose that $(x_i, y_i)$ is correctly classified by $f_{\theta(t)}$ in some time interval $t \in I$ and $\hat{x}_i(t)$ is a locally closest boundary point in the sense that for any $t \in I$, it is the local optimum of the following:*

$$\hat{x}_i(t) = \arg\min_{\hat{x}} \|\hat{x} - x_i\|_p \quad s.t. \quad \phi^{y_i}(\hat{x}, \theta(t)) = 0.$$

*Assume that in the time interval $I$: (a) $\hat{x}_i(t)$ is differentiable in $t$; (b) $\phi^{y_i}$ is differentiable at point $\hat{x}_i(t)$ and at the current parameter $\theta(t)$.*

**Proposition** (Closed-form expression of the speed $s(x_i, t)$)**.** *For $1 \leq p \leq \infty$ and under Assumption 8, define the (local) speed according to $\hat{x}_i(t)$ in Assumption 8 as $s(x_i, t) = \frac{d}{dt}\|\hat{x}_i(t) - x_i\|_p$, we have the following:*

$$s(x_i, t) = \frac{1}{\|\nabla_x \phi^{y_i}(\hat{x}_i(t), \theta(t))\|_q} \nabla_\theta \phi^{y_i}(\hat{x}_i(t), \theta(t)) \cdot \theta'(t)$$

*where $q$ satisfies that $1/q + 1/p = 1$. In particular, $q = 2$ when $p = 2$ and $q = 1$ when $p = \infty$.*

*Proof. Case 1: $1 \leq p < \infty$*

To compute $s(x_i, t) = \frac{d}{dt}\|\hat{x}_i(t) - x_i\|_2$, we need to characterize the curve of the closest boundary point $\hat{x}_i(t)$, where two key points stand out. First, $\hat{x}_i(t)$ is on the decision boundary $\Gamma_y(t)$ and thus $\phi^y(\hat{x}_i(t), \theta(t)) = 0$ for all $t \in I$. By taking the time derivative on both sides, we obtain the level set equation (Osher et al., 2004; Aghasi et al., 2011)

$$\nabla_x \phi^{y_i}(\hat{x}_i(t), \theta(t)) \cdot \hat{x}_i'(t) + \nabla_\theta \phi^{y_i}(\hat{x}_i(t), \theta(t)) \cdot \theta'(t) = 0 \tag{10}$$

Second, $\hat{x}_i(t)$ is the optimal solution of constrained optimization equation 3. Therefore, we have the following optimality condition:

$$\nabla_x \phi^{y_i}(\hat{x}_i(t), \theta(t)) + \lambda(t) \nabla_x \|\hat{x}_i(t) - x_i\|_p = 0 \tag{11}$$

Since $x_i$ is correctly classified, $\phi^{y_i}(x_i) > 0$. Since $\hat{x}_i(t)$ is the closest point to $x_i$ whose $\phi^{y_i}$ value is zero, $\lambda(t) > 0$. By taking the $L_q$ norm in Equation equation 11 and using Lemma 6, we obtain that $\lambda(t) = \|\nabla_x \phi^{y_i}(\hat{x}_i(t), \theta(t))\|_q$.

Now, we derive $s(x_i, t)$ as follows:

$$
\begin{aligned}
s(x_i, t) &= \frac{d}{dt}\|\hat{x}_i(t) - x_i\|_p \\
&= \nabla_x\|\hat{x}_i(t) - x_i\|_p \cdot \hat{x}_i'(t) \\
&= -\frac{1}{\lambda}\nabla_x\phi^{y_i}(\hat{x}_i(t), \theta(t)) \cdot \hat{x}_i'(t) \quad \text{(By the optimality condition equation 11)} \\
&= \frac{1}{\lambda}\nabla_\theta\phi^{y_i}(\hat{x}_i(t), \theta(t)) \cdot \theta'(t) \quad \text{(By the level set equation equation 10)} \\
&= \frac{\nabla_\theta\phi^{y_i}(\hat{x}_i(t), \theta(t)) \cdot \theta'(t)}{\|\nabla_x\phi^y(\hat{x}_i(t), \theta(t))\|_q}
\end{aligned}
$$

*Case 2:* $p = \infty$

Note that since $L_\infty$ is not differentiable, the optimality condition in Equation equation 11 does not hold anymore.

Denote the $j$-th component of $\hat{x}_i(t)$ and $x_i$ as $\hat{x}_{ij}(t)$ and $x_{ij}$. Let $\mathcal{J} = \{j : |\hat{x}_{ij}(t) - x_{ij}| = \|\hat{x}_i(t) - x_i\|_\infty\}$. By Lemma 7, $s(x_i, t) = \frac{d}{dt}|\hat{x}_{ij}(t) - x_{ij}| = \hat{x}_{ij}'(t)\,\mathrm{sign}(\hat{x}_{ij}(t) - x_{ij}) = -\hat{x}_{ij}'(t)\,\mathrm{sign}(\nabla_x\phi^{y_i}(\hat{x}_i)_j)$ for all $j \in \mathcal{J}$. Therefore, by Equation equation 10 and Lemma 7

$$
\begin{aligned}
-\nabla_\theta\phi^{y_i}(\hat{x}_i(t), \theta(t)) \cdot \theta'(t) &= \nabla_x\phi^{y_i}(\hat{x}) \cdot \hat{x}_i'(t) \\
&= \sum_{j\in\mathcal{J}} \nabla_x\phi^{y_i}(\hat{x}_i)_j \cdot \hat{x}_{ij}'(t) \\
&= \sum_{j\in\mathcal{J}} -\nabla_x\phi^{y_i}(\hat{x}_i)_j \cdot \frac{s(x_i, t)}{\mathrm{sign}(\nabla_x\phi^y(\hat{x}_i)_j)} \\
&= -\sum_{j\in\mathcal{J}} |\nabla_x\phi^{y_i}(\hat{x}_i)_j| \cdot s(x_i, t)
\end{aligned}
$$

Therefore, $s(x_i, t) = \frac{\nabla_\theta\phi^{y_i}(\hat{x}_i(t), \theta(t))\cdot\theta'(t)}{\sum_{j\in\mathcal{J}}|\nabla_x\phi^{y_i}(\hat{x}_i)_j|} = \frac{\nabla_\theta\phi^{y_i}(\hat{x}_i(t), \theta(t))\cdot\theta'(t)}{\|\nabla_x\phi^{y_i}(\hat{x}_i)\|_1}$, where the last equality follows from Lemma 7 that the components of $\nabla_x\phi^{y_i}(\hat{x}_i)$ are zeros if they are not in $\mathcal{J}$.

$\square$

As an corollary of the proposition we prove above, we can obtain the closed-form expression for the gradient of margin (or the gradient of any smooth function of the margin) as follows:

**Theorem** (Closed-form expression of $\nabla_\theta h(R_\theta(x_i))$)**.** *For* $1 \le p \le \infty$,

$$
\nabla_\theta h(R_\theta(x_i)) = \frac{h'(R_\theta(x_i))}{\|\nabla_x\phi^{y_i}(\hat{x}_i, \theta)\|_q}\nabla_\theta\phi^{y_i}(\hat{x}_i, \theta)
$$

*where $q$ satisfies that $1/q + 1/p = 1$.*

*Proof.* In continuous time we consider $h(R(x_i, t))$ (or more rigorously, $h(R(x_i, \theta(t)))$) and its time derivative. We use the following relationship between the gradient and the time derivative, where $\theta'(t)$ can be any update rule:

$$
\frac{d}{dt}h(R(x_i, t)) = \nabla_\theta h(R(x_i, t)) \cdot \theta'(t)
$$

On the other hand:

$$
\begin{aligned}
\frac{d}{dt}h(R(x_i, t)) &= h'(R(x_i, t))\frac{d}{dt}R(x_i, t) \\
&= h'(R(x_i, t))s(x_i, t) \\
&= \frac{h'(R(x_i))}{\|\nabla_x\phi^{y_i}(\hat{x}_i(t), \theta(t))\|_q}\nabla_\theta\phi^{y_i}(\hat{x}_i(t), \theta(t)) \cdot \theta'(t)
\end{aligned}
$$

where the last equality uses the closed-form expression for the speed $s(x_i, t)$.

Therefore we have that for any $\theta'(t)$, $\nabla_\theta h(R(x_i, t)) \cdot \theta'(t) = \frac{h'(R(x_i))}{\|\nabla_x \phi^{y_i}(\hat{x}_i(t), \theta(t))\|_q} \nabla_\theta \phi^{y_i}(\hat{x}_i(t), \theta(t)) \cdot \theta'(t)$. We conclude that $\nabla_\theta h(R_\theta(x_i)) = \frac{h'(R_\theta(x_i))}{\|\nabla_x \phi^{y_i}(\hat{x}_i, \theta)\|_q} \nabla_\theta \phi^{y_i}(\hat{x}_i, \theta)$.

$\square$

### DISCUSSIONS ON THE ASSUMPTIONS

Assumption 8 has several points that need to be explained further.

First, we only require that $\hat{x}_i(t)$ is a local closest boundary point. This is important because in practice when an algorithm for searching the closest boundary point is used (e.g. FAB), a local solution is the best one can hope for due to the non-convex nature of the optimization problem. When $\hat{x}_i(t)$ is a local solution, the speed should be interpreted as how fast the distance changes around that local solution. In this case, although $\hat{x}_i$ is not the globally closest adversarial example, the local speed around $\hat{x}_i$ still has much information on the relative movement of the decision boundary w.r.t. $x_i$, especially when the distance $\|\hat{x}_i - x_i\|$ is relatively small and the input space is a high-dimensional space (e.g. pixel space).

Second, we require that $\hat{x}_i(t)$ is a differentiable curve in time interval $I$. Note that if we constrain $\hat{x}_i(t)$ to be the global closest boundary point, $\hat{x}_i$ is unlikely to be differential for a large time interval $I$, especially in high dimensional space. This is because as the decision boundary moves. the closest point might switch from one point to another point that is very far away abruptly. Fortunately, this problem is alleviated because our derived closed-form expression still works when $\hat{x}_i(t)$ is a locally closest boundary point. Note that however, due to the *topological change* of the decision boundary, it still can happen that $\hat{x}_i(t)$ stops existing (and thus stops being differentiable) at some time point, when typically the speed will go to infinity. From a mathematical point of view, this is related to *shock* in partial differential equation (PDE) theories. More exploration on this is left to future work. In this work, we only consider the speed of the decision boundary at each discrete time step.

## C   COMPUTATION OF THE EXACT AND SOFT CLOSEST BOUNDARY POINT

Either computing the speed of the decision boundary or using DyART to directly optimize a function of margins requires the computation of the closest boundary point $\hat{x}$ (or the closest soft boundary point $\hat{x}^{\text{soft}}$), where we omit the subscript $i$ in this section. As discussed in Appendix B, it suffices to find the locally closest (soft) boundary point in order for the closed-form expression 4 and expression 6 for the speed and the gradient of margin to be valid.

### C.1   CLOSEST BOUNDARY POINT

In this section, we will explain how to check the quality of the found $\hat{x}$ for the constrained optimization problem 3 in practice. We will also give a simple analysis on how FAB(Croce & Hein, 2020a), the algorithm we use in our implementation, solves the problem 3 in practice. We include both $p = 2$ and $p = \infty$ although in our work, only $p = \infty$ is used. We discuss both of them in order to highlight the difference in checking optimality conditions for smooth ($p = 2$) and non-smooth norm ($p = \infty$).

The key points of analyzing $\hat{x}$ are that $\phi^y(\hat{x}) = 0$ and the KKT conditions of problem 3.

*Case 1: $p = 2$*

In this case, the KKT condition is given by $\nabla_x \phi^y(\hat{x}) + \lambda(\hat{x} - x) = 0$ for some $\lambda > 0$ (since $\phi^y(x) > 0$). In other words, $\frac{\nabla_x \phi^y(\hat{x})}{\|\nabla_x \phi^y(\hat{x})\|} \cdot \frac{x - \hat{x}}{\|x - \hat{x}\|} = 1$. In practice, we check the following two conditions (a) $|\phi(\hat{x})| \leq 0.1$; (b) $\frac{\nabla_x \phi^y(\hat{x})}{\|\nabla_x \phi^y(\hat{x})\|} \cdot \frac{x - \hat{x}}{\|x - \hat{x}\|} > 0.8$. We observe in our experiments that FAB can find high-quality closest boundary points for over 90% of the correctly classified data points.

*Case 2: $p = \infty$*

In this case, we consider the optimality condition given in Lemma 7 of Appendix B. Denote $\#B$ the number of points in a set $B$. Using the notation $\mathcal{J} = \{j : |\hat{x}_j - x_j| = \|\hat{x} - x\|_\infty\}$ and $\mathcal{J}^C$ the complement set of $\mathcal{J}$, we check the following conditions in practice: (a) $|\phi(\hat{x})| \leq 0.1$; (b) $\frac{\#\{j \in \mathcal{J} : \nabla_x \phi^y(\hat{x})_j (\hat{x}_j - x_j) \leq 0\}}{\#\mathcal{J}} > 0.9$; (c) $\frac{\#\{k \notin J : |\nabla_x \phi^y(\hat{x})_k| < 0.1\}}{\#\mathcal{J}^C} > 0.8$. Note the unlike $p = 2$, the optimality conditions for $p = \infty$ are on each coordinate of $\hat{x}$, which is more difficult to satisfy in practice. We observe in our experiments that FAB with 100 iterations can find high-quality closest boundary points for about 85% of the correctly classified points. However, when only 20 iterations are used, condition (3) is barely satisfied for all of the found boundary points (the first two conditions are still satisfied).

In our visualizations of dynamics of the decision boundary for AT in Section 4.2, we use 100 iterations for FAB and only use high-quality closest boundary points, so that the visualization results are relatively accurate.

### C.2   CLOSEST SOFT BOUNDARY POINT

**Adapt FAB for soft decision boundary.**   In DyART, the closest point $\hat{x}^{\text{soft}}$ on the soft decision boundary is used. To find $\hat{x}^{\text{soft}}$, we adapt the FAB method. The original FAB method aims to find the closest point on the exact decision boundary. In particular, FAB forms linear approximations for decision boundary between the ground truth class and every other classes. The only adaptation we do on the FAB method is that now FAB only forms *one* linear approximation for the soft decision boundary of the ground truth class. This is because we use the smoothed max operator in the soft logit margin, and there is no concept of the 'decision boundary between the ground truth class and another class' anymore.

**Computational efficiency.**   By using the soft decision boundary, every iteration of FAB only requires one linear approximation of the soft decision boundary, which cost one back-propagation. In contrast, the original FAB which aims to find the closest boundary point on the exact decision boundary costs $K$ back-propagation at each iteration, where $K$ is the number of classes. Therefore, using the soft decision boundary is more efficient and is used in our proposed robust training method DyART.

**Local optimality condition.** The procedure of checking optimality condition is similar to the one in the last section. Denote $\#B$ the number of points in a set $B$. Using the notation $\mathcal{J} = \{j : |\hat{x}_j - x_j| = \|\hat{x} - x\|_\infty\}$ and $\mathcal{J}^C$ the complement set of $\mathcal{J}$, we check the following conditions in practice: (a) $|\phi(\hat{x})| \leq 0.1$; (b) $\frac{\#\{j\in\mathcal{J}:\nabla_x\phi^y(\hat{x})_j(\hat{x}_j-x_j)\leq 0\}}{\#\mathcal{J}} > 0.9$; We find that when the temperature $\beta$ is relatively large (we use $\beta = 5$ in all of our experiments) and 20 iterations is used, $95\%$ of the soft boundary point found for the correctly classified points satisfy these two conditions. During training, we only use these higher quality points and discard the rest of the boundary points that do not satisfy these two conditions. Note that we do not consider the third condition (c) $\frac{\#\{k\notin J:|\nabla_x\phi^y(\hat{x})_k|<0.1\}}{\#\mathcal{J}^C} > 0.8$. This is because condition (c) cannot be satisfied unless a very large iteration number is used, which is computationally prohibitive for robust training.

Experimentally `DyART` achieves improved robustness over baseline methods, indicating that the closest soft boundary points used by `DyART` are indeed useful for robust training. Designing faster and more reliable methods to solve the constrained optimization problem 3 is left for future work.

# D  EXPERIMENTS

In this section, we provide the details of experimental settings and further results of `DyART` using various choices of hyperparameters. In addition, we provide experimental results when using additional data from the generated models. We also provide further analysis on the decision boundary dynamics.

## D.1  DETAILED EXPERIMENTAL SETTINGS

**Architectures and training settings.** In all experiments on the CIFAR-10 dataset, we use the Wide Residual Network (Zagoruyko & Komodakis, 2016) with depth 28 and width factor 10 (WRN-28-10) with Swish activation function (Ramachandran et al., 2017). On the Tiny-ImageNet dataset, we use pre-activation ResNet-18 (He et al., 2016b). In all experiments, we use stochastic weight averaging (Izmailov et al., 2018) with a decay rate of $0.995$ as in prior work (Gowal et al., 2020; Chen et al., 2020). All models are trained using stochastic gradient descent with momentum $0.9$ and weight decay $0.0005$. We use a cosine learning rate schedule (Loshchilov & Hutter, 2016) without restarts where the initial learning rate is set to $0.1$ for baselines. To alleviate robust overfitting (Rice et al., 2020), we compute the robust and clean accuracy at every epoch on a validation set of size 1024 using projected gradient descent (PGD) attacks with 20 steps using margin loss function. All experiments are run on NVIDIA GeForce RTX 2080 Ti GPU.

**Normalization layers** We consider two types of normalization layer in WRN-28-10 and ResNet-18, which are Batch Normalization (BN, used in their original architecture design) and Group Normalization (GN). When using GN, the decision boundaries are the same during training and evaluation, which is consistent with our theoretical analysis on the decision boundary dynamics. In the following sections, we will show the robustness performance on both cases: WRN-28-10 and ResNet-18 with BN and GN. We find that when applying `DyART` on original WRN-28-10 and ResNet-18 with BN, gradient clipping needs to be applied in order to learn the BN parameters stably. We apply gradient clipping with norm threshold $0.1$ for experiments for CIFAR-10 on WRN-28-10 with BN and apply gradient clipping with norm threshold 1 for Tiny-ImageNet on ResNet-18 with BN. For experiments on architectures with GN, we do not apply gradient clipping. Note that for all experiments of computing speed and margins for interpretation (Section 4 and Section 6.2), we use the ResNet-18 with GN.

**Additional training settings** For experiments with Group Normalization, models are run for 100 epochs on both datasets. For `DyART` on Tiny-ImageNet, we use the cosine learning rate schedule with initial learning rate $0.05$ and on CIFAR-10, the learning rate begins at $0.1$ and is decayed by a factor of 10 at the 50th and 75th epoch. For experiments with Batch Normalization, models are run for 200 epochs on CIFAR-10 and 100 epochs on Tiny-ImageNet. For `DyART` on both datasets, we use a cosine learning rate schedule (Loshchilov & Hutter, 2016) without restarts where the initial learning rate is set to $0.1$, which is the same as the baselines.

**Compared baselines and their hyperparameters.** In all experiments we consider the $\ell_\infty$ perturbation setting. On CIFAR-10, the baseline defense methods include: (1) standard adversarial training (AT) (Madry et al., 2017) which trains on the worst case adversarial examples generated by 10-step PGD (PGD-10) on the cross-entropy loss. The perturbation bound is $\frac{8}{255}$ and the step size of PGD is $\frac{2}{255}$; the training setting follows Rice et al.[1] (Rice et al., 2020). (2) TRADES [2](Zhang et al., 2019) which trades off between the clean and robust accuracy. The perturbation bound is $\frac{8}{255}$ with the step size of PGD-10 $0.007$. The regularization constant beta (or 1/lambda) is set to 6. (3) MMA [3] (Ding et al., 2020) which trains on the closest adversarial examples (closest boundary points) with uniform weights. The MaxEps is set to $\frac{32}{255}$. (4) GAIRAT [4] (Zhang et al., 2021) which reweights adversarial examples using the least perturbation steps. The perturbation bound is $\frac{8}{255}$ with step

---

[1]Robust Overfitting's Github

[2]TRADES's Github

[3]MMA's Github

[4]GAIRAT's Github

size 0.007 using PGD-10 and the 'tanh' weight assignment function is used. (5) MAIL [5] (Liu et al., 2021) which reweights adversarial examples using margin value. We choose its combination with TRADES (MAIL-TRADES) which provides better robustness performance than combining with AT (MAIL-AT). Its hyperparamters beta, bias and slope are set to $5.0, -1.5$ and $1.0$, respectively. (6) AWP [6] (Wu et al., 2020) which adversarially perturbs both inputs and model parameters. (7) FAT [7] (Zhang et al., 2020b) that exploits friendly adversarial data, where the perturbation bound is set to $\frac{8}{255}$. (8) MART (Wang et al., 2019) which explicitly differentiates the mis-classified and correctly classified examples. On Tiny-ImageNet, we compare with AT, TRADES, and MART whose hyperparameter settings are available for this dataset. We follow the PyTorch implementation of (Gowal et al., 2020; Rebuffi et al., 2021) [8] for AT, TRADES and MART for both datasets.

**Evaluation details.** We evaluate DyART and the baselines under $\ell_\infty$ norm constrained perturbations. The final robust accuracy is reported on AutoAttack (AA) (Croce & Hein, 2020b), which uses an ensemble of selected strong attacks. For all methods, we choose the hyperparameters to achieve the best robust accuracy under the commonly used perturbation bound $\epsilon = \frac{8}{255}$. To fully compare the robustness performance among different methods, we report the robust accuracy under four additional perturbation bounds: $\frac{2}{255}, \frac{4}{255}, \frac{12}{255}$ and $\frac{16}{255}$.

**Per-sample gradient** For computing the speed of the decision boundary in Section 4.2 and Section 6.2, we need to compute the per-sample gradient $\nabla_\theta \phi^{y_i}(\hat{x}_i, \theta)$ for every correctly classified point $x_i$. We use the Opacus package (Yousefpour et al., 2021) for computing per-sample gradients in parallel. Also, another reason why we replace BN with GN is because Opacus does not support BN for computing per-sample gradients. Although using this package will increase the memory usage, it is worth mentioning that during robust training, DyART does not need to compute the per-sample gradient and thus does not have the excessive memory issue. Per-sample gradients are only collected for computing speed, which is for interpretation of dynamics of different methods and not for robust training.

## D.2 HYPERPARAMETER SENSITIVITY EXPERIMENTS

In this section, we present the robustness performance of DyART under different hyperparameter settings. We first show the results for architectures using Group Normalization (note that in Section 6.1 we use the original architectures using Batch Normalization) and analyze the effect of different hyperparameters. We then demonstrate more ablation experiments for architectures using Batch Normalization used in Section 6.1.

| Defense | Clean | $\epsilon = \frac{2}{255}$ | $\epsilon = \frac{4}{255}$ | $\epsilon = \frac{8}{255}$ | $\epsilon = \frac{12}{255}$ | $\epsilon = \frac{16}{255}$ |
|---|---|---|---|---|---|---|
| AT | $85.36 \pm 0.17$ | $77.16 \pm 0.29$ | $67.84 \pm 0.24$ | $46.27 \pm 0.19$ | $26.62 \pm 0.18$ | $12.40 \pm 0.12$ |
| TRADES | $84.67 \pm 0.24$ | $77.72 \pm 0.18$ | $69.38 \pm 0.15$ | $49.29 \pm 0.15$ | $30.25 \pm 0.17$ | $16.42 \pm 0.18$ |
| MART | $81.02 \pm 0.17$ | $73.04 \pm 0.21$ | $64.94 \pm 0.22$ | $48.06 \pm 0.20$ | $30.61 \pm 0.13$ | $16.42 \pm 0.09$ |
| MMA | $85.52 \pm 0.36$ | $74.78 \pm 0.42$ | $62.21 \pm 0.39$ | $38.61 \pm 0.47$ | $22.13 \pm 0.29$ | $9.95 \pm 0.20$ |
| GAIRAT | $83.72 \pm 0.27$ | $73.87 \pm 0.33$ | $61.7 \pm 0.15$ | $37.77 \pm 0.21$ | $18.87 \pm 0.15$ | $8.1 \pm 0.11$ |
| MAIL-TRADES | $84.48 \pm 0.22$ | $77.18 \pm 0.26$ | $68.20 \pm 0.31$ | $48.64 \pm 0.12$ | $29.87 \pm 0.11$ | $15.62 \pm 0.16$ |
| FAT-TRADES | $\mathbf{86.58} \pm 0.25$ | $\mathbf{78.96} \pm 0.17$ | $69.54 \pm 0.12$ | $48.07 \pm 0.19$ | $27.66 \pm 0.12$ | $13.22 \pm 0.23$ |
| DyART | $85.64 \pm 0.10$ | $78.20 \pm 0.16$ | $\mathbf{69.59} \pm 0.19$ | $\mathbf{50.03} \pm 0.16$ | $\mathbf{30.87} \pm 0.20$ | $\mathbf{16.55} \pm 0.12$ |

**Table 3:** Clean and robust accuracy on CIFAR-10 under AA with different perturbation sizes on WRN-28-10 with Group Normalization. The hyperparameters for DyART is $\alpha = 8, r_0 = \frac{16}{255}, \lambda = 400$.

**Overall performance of DyART on architectures with GN** In Table 3 and Table 4, the overall comparison between DyART and baselines are demonstrated. Overall on both datasets, under four out of five perturbation bounds, DyART achieves the best robustness performance. This indicates the superiority of DyART in increasing margins. **(1)** Specifically, on CIFAR-10, DyART achieves the highest clean accuracy as well as robust accuracy under all perturbation bounds among all baselines except FAT-TRADES. (1a) Since FAT-TRADES prevents the model from learning on

---

[5] MAIL's github

[6] AWP's github

[7] FAT's github

[8] UncoveringATLimits's Github

| Defense | Clean | $\epsilon = \frac{2}{255}$ | $\epsilon = \frac{4}{255}$ | $\epsilon = \frac{8}{255}$ | $\epsilon = \frac{12}{255}$ | $\epsilon = \frac{16}{255}$ |
|---------|-------|------------|------------|------------|-------------|-------------|
| AT | $43.76 \pm 0.53$ | $35.54 \pm 0.36$ | $28.20 \pm 0.21$ | $16.92 \pm 0.24$ | $9.34 \pm 0.18$ | $4.75 \pm 0.14$ |
| TRADES | $46.56 \pm 0.29$ | $37.23 \pm 0.17$ | $28.68 \pm 0.19$ | $16.20 \pm 0.21$ | $8.38 \pm 0.10$ | $4.23 \pm 0.06$ |
| MART | $38.74 \pm 0.42$ | $32.18 \pm 0.74$ | $26.08 \pm 0.31$ | $16.90 \pm 0.26$ | $10.14 \pm 0.22$ | $\mathbf{6.10} \pm 0.19$ |
| DyART | $\mathbf{47.67} \pm 0.15$ | $\mathbf{38.19} \pm 0.18$ | $\mathbf{29.59} \pm 0.14$ | $\mathbf{17.79} \pm 0.18$ | $\mathbf{10.24} \pm 0.13$ | $5.41 \pm 0.11$ |

**Table 4:** Clean and robust accuracy on Tiny-ImageNet under AA with different perturbation sizes on ResNet-18 with Group Normalization. The hyperparameters for DyART is $\alpha = 3, r_0 = \frac{20}{255}, \lambda = 500$.

highly adversarial data in order to keep clean accuracy high, it achieves the best clean accuracy and robustness under a very small perturbation bound $\frac{2}{255}$. However, its performance on larger perturbation bounds is inadequate. (1b) Thanks to directly operating on margins in the input space and encourage robustness improvement on points with smaller margins, DyART performs better than GAIRAT and MAIL-TRADES which use indirect approximations of the margins. **(2)** On Tiny-ImageNet, DyART achieves the best clean accuracy and the best robust accuracy under all perturbation bounds except the largest $\frac{16}{255}$. Although MART is the most robust under $\frac{16}{255}$, it has much lower clean accuracy (8.93% lower than DyART) and worse robustness under smaller perturbation bounds.

**Hyperparameters of DyART** In this paper, we use the cost function of the form $h(R) = \frac{1}{\alpha} \exp(-\alpha R)$ when $R < r_0$ and $h(R) = 0$ otherwise. We present results under different decay strengh $\alpha > 0$, margin threshold $r_0$ as well as regularization constant $\lambda$ for the robustness loss.

**Performance results.** The evaluation results on CIFAR-10 and Tiny-ImageNet with Group Normalization are shown in Table 5 and Table 6, respectively. We analyze the effects of hyperparameters as follows.

**(1)** Effect of $\alpha$: Larger $\alpha$ corresponds to a cost function $h(\cdot)$ that decays faster, and therefore prioritize improvement on even smaller margins. Therefore, it should be expected that larger $\alpha$ leads to higher clean accuracy and higher robust accuracy under smaller perturbation sizes, and results in lower robust accuracy under larger perturbation sizes. For example, on CIFAR-10, when $\alpha = 5$ is increased to $\alpha = 8$ when $r_0 = \frac{16}{255}, \lambda = 400$, the clean accuracy as well as the robust accuracy under $\epsilon = \frac{2}{255}, \frac{4}{255}$ and $\frac{8}{255}$ improves, while the robust accuracy under larger $\epsilon$ gets lower. The same patterns can also be observed on Tiny-ImageNet, for example, when $\alpha = 8$ is increased to $\alpha = 10$ when $r_0 = \frac{20}{255}, \lambda = 1000$.

**(2)** Effect of $r_0$: $r_0$ is from preventing DyART from training boundary points that are too far away from clean data points. Therefore, it should be expected that training on smaller $r_0$ tends to increase the clean accuracy and the robust accuracy under relatively small perturbation sizes. Indeed, on Tiny-ImageNet, when $r_0 = \frac{24}{255}$ is decreased to $r_0 = \frac{20}{255}$ when $\alpha = 10$ and $\lambda = 1000$, we can observe that the clean accuracy as well as robust accuracy under $\epsilon = \frac{2}{255}$ increases but the robust accuracy under larger perturbation sizes $\epsilon = \frac{12}{255}$ and $\frac{16}{255}$ decreases.

**(3)**: Effect of robust loss constant $\lambda$: A larger $\lambda$ tends to increase the robustness of the model (in particular, the robust accuracy under relatively larger perturbation sizes) while decrease the clean accuracy and the robust accuracy under relatively small perturbation sizes. For example, on Tiny-ImageNet, when $\lambda = 800$ is increased to $\lambda = 1000$ when $\alpha = 10$ and $r_0 = \frac{20}{255}$, the clean accuracy and robust accuracy under relatively small $\epsilon = \frac{2}{255}, \frac{4}{255}$ drops but the robust accuracy under larger perturbation sizes increase.

**(4)** Effect of the burn-in period: A burn-in period of natural training is necessary for DyART since its robust loss function depends on the closest boundary points, which can only be found on correctly classified points. That is, DyART requires a descent initial clean accuracy. In our experiments, we find that the learning rate of the burn-in period is important: DyART will train successfully if the learning rate of the burn-in period is relatively large (e.g. 0.1 for CIFAR-10 and Tiny-ImageNet). However, when the learning rate is small (such as 0.001), DyART sometimes drives the clean accuracy to be very low at first, and fails to train. Our suggestion is to use a larger learning rate to obtain a naturally pretrained model.

| Defense | Clean | $\epsilon = \frac{2}{255}$ | $\epsilon = \frac{4}{255}$ | $\epsilon = \frac{8}{255}$ | $\epsilon = \frac{12}{255}$ | $\epsilon = \frac{16}{255}$ |
|---|---|---|---|---|---|---|
| AT | $85.36 \pm 0.17$ | $77.16 \pm 0.29$ | $67.84 \pm 0.24$ | $46.27 \pm 0.19$ | $26.62 \pm 0.18$ | $12.40 \pm 0.12$ |
| TRADES | $84.67 \pm 0.24$ | $77.72 \pm 0.18$ | $69.38 \pm 0.15$ | $49.29 \pm 0.15$ | $30.25 \pm 0.17$ | $16.42 \pm 0.18$ |
| MART | $81.02 \pm 0.17$ | $73.04 \pm 0.21$ | $64.94 \pm 0.22$ | $48.06 \pm 0.20$ | $30.61 \pm 0.13$ | $16.42 \pm 0.09$ |
| MMA | $85.52 \pm 0.36$ | $74.78 \pm 0.42$ | $62.21 \pm 0.39$ | $38.61 \pm 0.47$ | $22.13 \pm 0.29$ | $9.95 \pm 0.20$ |
| GAIRAT | $83.72 \pm 0.27$ | $73.87 \pm 0.33$ | $61.7 \pm 0.15$ | $37.77 \pm 0.21$ | $18.87 \pm 0.15$ | $8.1 \pm 0.11$ |
| MAIL-TRADES | $84.48 \pm 0.22$ | $77.18 \pm 0.26$ | $68.20 \pm 0.31$ | $48.64 \pm 0.12$ | $29.87 \pm 0.11$ | $15.62 \pm 0.16$ |
| FAT-TRADES | $\mathbf{86.58} \pm 0.25$ | $\mathbf{78.96} \pm 0.17$ | $69.54 \pm 0.12$ | $48.07 \pm 0.19$ | $27.66 \pm 0.12$ | $13.22 \pm 0.23$ |
| $\alpha = 10, r_0 = \frac{20}{255}, \lambda = 400$ | $84.17 \pm 0.12$ | $76.85 \pm 0.15$ | $68.47 \pm 0.15$ | $49.41 \pm 0.20$ | $32.07 \pm 0.17$ | $18.72 \pm 0.10$ |
| $\alpha = 5, r_0 = \frac{20}{255}, \lambda = 300$ | $83.33 \pm 0.19$ | $75.96 \pm 0.24$ | $67.82 \pm 0.22$ | $49.55 \pm 0.24$ | $\mathbf{32.29} \pm 0.14$ | $\mathbf{19.16} \pm 0.15$ |
| $\alpha = 8, r_0 = \frac{16}{255}, \lambda = 400$ | $85.64 \pm 0.10$ | $78.20 \pm 0.16$ | $\mathbf{69.59} \pm 0.19$ | $\mathbf{50.03} \pm 0.16$ | $30.87 \pm 0.20$ | $16.55 \pm 0.12$ |
| $\alpha = 5, r_0 = \frac{16}{255}, \lambda = 400$ | $85.05 \pm 0.14$ | $77.92 \pm 0.21$ | $69.00 \pm 0.14$ | $49.60 \pm 0.12$ | $30.78 \pm 0.14$ | $17.06 \pm 0.09$ |
| $\alpha = 0, r_0 = \frac{16}{255}, \lambda = 400$ | $83.85 \pm 0.23$ | $76.77 \pm 0.20$ | $68.26 \pm 0.15$ | $49.65 \pm 0.18$ | $31.72 \pm 0.21$ | $17.76 \pm 0.18$ |

**Table 5:** Clean and robust accuracy on CIFAR-10 under AA with different perturbation bounds on WRN-28-10 with Group Normalization. The results on different sets of hyperparameters for DyART starts from the eighth row.

| Defense | Clean | $\epsilon = \frac{2}{255}$ | $\epsilon = \frac{4}{255}$ | $\epsilon = \frac{8}{255}$ | $\epsilon = \frac{12}{255}$ | $\epsilon = \frac{16}{255}$ |
|---|---|---|---|---|---|---|
| AT | $43.76 \pm 0.53$ | $35.54 \pm 0.36$ | $28.20 \pm 0.21$ | $16.92 \pm 0.24$ | $9.34 \pm 0.18$ | $4.75 \pm 0.14$ |
| TRADES | $46.56 \pm 0.29$ | $37.23 \pm 0.17$ | $28.68 \pm 0.19$ | $16.20 \pm 0.21$ | $8.38 \pm 0.10$ | $4.23 \pm 0.06$ |
| MART | $38.74 \pm 0.42$ | $32.18 \pm 0.74$ | $26.08 \pm 0.31$ | $16.90 \pm 0.26$ | $10.14 \pm 0.22$ | $\mathbf{6.10} \pm 0.19$ |
| $\alpha = 10, r_0 = \frac{32}{255}, \lambda = 500$ | $\mathbf{48.98} \pm 0.24$ | $\mathbf{38.38} \pm 0.32$ | $\mathbf{29.76} \pm 0.22$ | $17.30 \pm 0.19$ | $9.87 \pm 0.11$ | $5.19 \pm 0.15$ |
| $\alpha = 10, r_0 = \frac{24}{255}, \lambda = 1000$ | $45.27 \pm 0.19$ | $36.58 \pm 0.52$ | $29.03 \pm 0.32$ | $17.35 \pm 0.24$ | $10.03 \pm 0.18$ | $5.61 \pm 0.16$ |
| $\alpha = 10, r_0 = \frac{20}{255}, \lambda = 1000$ | $46.37 \pm 0.26$ | $37.43 \pm 0.32$ | $29.01 \pm 0.19$ | $17.61 \pm 0.20$ | $9.91 \pm 0.18$ | $5.27 \pm 0.14$ |
| $\alpha = 10, r_0 = \frac{20}{255}, \lambda = 800$ | $47.09 \pm 0.22$ | $38.04 \pm 0.12$ | $29.55 \pm 0.17$ | $17.22 \pm 0.15$ | $9.59 \pm 0.20$ | $5.08 \pm 0.11$ |
| $\alpha = 8, r_0 = \frac{20}{255}, \lambda = 1000$ | $45.69 \pm 0.17$ | $36.74 \pm 0.20$ | $28.57 \pm 0.12$ | $17.31 \pm 0.21$ | $10.13 \pm 0.15$ | $5.19 \pm 0.16$ |
| $\alpha = 5, r_0 = \frac{20}{255}, \lambda = 800$ | $45.61 \pm 0.14$ | $36.87 \pm 0.16$ | $29.01 \pm 0.19$ | $17.58 \pm 0.10$ | $\mathbf{10.38} \pm 0.16$ | $5.33 \pm 0.09$ |
| $\alpha = 3, r_0 = \frac{20}{255}, \lambda = 500$ | $47.67 \pm 0.15$ | $38.19 \pm 0.18$ | $29.59 \pm 0.14$ | $\mathbf{17.79} \pm 0.18$ | $10.24 \pm 0.13$ | $5.41 \pm 0.11$ |
| $\alpha = 3, r_0 = \frac{16}{255}, \lambda = 1000$ | $45.27 \pm 0.20$ | $36.71 \pm 0.16$ | $28.74 \pm 0.20$ | $17.40 \pm 0.16$ | $9.80 \pm 0.13$ | $4.92 \pm 0.13$ |

**Table 6:** Clean and robust accuracy on Tiny-ImageNet under AA with different perturbation bounds on ResNet-18 with Group Normalization. The results on different sets of hyperparameters for DyART starts from the fourth row.

**More ablation on architectures with BN**  In Table 7 and Table 8, we demonstrate results for more hyperparamter settings for experiments with Batch Normalization in Section 6.1. The role of each hyperparameter is similar to the GN case.

| Defense | Clean | $\epsilon=\frac{2}{255}$ | $\epsilon=\frac{4}{255}$ | $\epsilon=\frac{8}{255}$ | $\epsilon=\frac{12}{255}$ | $\epsilon=\frac{16}{255}$ |
|---|---|---|---|---|---|---|
| AT | $85.65 \pm 0.25$ | $79.08 \pm 0.12$ | $71.24 \pm 0.28$ | $53.20 \pm 0.16$ | $32.94 \pm 0.32$ | $16.12 \pm 0.23$ |
| TRADES | $82.92 \pm 0.30$ | $77.69 \pm 0.16$ | $70.68 \pm 0.15$ | $54.28 \pm 0.19$ | $36.65 \pm 0.24$ | $21.59 \pm 0.31$ |
| MART | $83.37 \pm 0.25$ | $76.58 \pm 0.24$ | $70.19 \pm 0.18$ | $52.91 \pm 0.24$ | $35.16 \pm 0.13$ | $18.80 \pm 0.14$ |
| MMA | $83.22 \pm 0.38$ | $74.24 \pm 0.52$ | $64.42 \pm 0.29$ | $44.02 \pm 0.33$ | $26.45 \pm 0.21$ | $13.78 \pm 0.25$ |
| GAIRAT | $\mathbf{86.59} \pm 0.31$ | $76.72 \pm 0.28$ | $64.64 \pm 0.25$ | $38.16 \pm 0.32$ | $19.01 \pm 0.18$ | $7.55 \pm 0.17$ |
| AWP | $84.27 \pm 0.19$ | $78.33 \pm 0.21$ | $70.82 \pm 0.26$ | $53.92 \pm 0.17$ | $35.24 \pm 0.26$ | $20.40 \pm 0.14$ |
| $\alpha=0, r_0=\frac{16}{255}, \lambda=1000$ | $85.10 \pm 0.24$ | $78.68 \pm 0.18$ | $71.67 \pm 0.28$ | $54.78 \pm 0.21$ | $36.26 \pm 0.24$ | $21.55 \pm 0.16$ |
| $\alpha=3, r_0=\frac{16}{255}, \lambda=1000$ | $85.55 \pm 0.24$ | $79.21 \pm 0.14$ | $71.73 \pm 0.18$ | $54.69 \pm 0.14$ | $35.74 \pm 0.25$ | $20.79 \pm 0.18$ |
| $\alpha=3, r_0=\frac{16}{255}, \lambda=1500$ | $85.34 \pm 0.19$ | $78.97 \pm 0.21$ | $71.82 \pm 0.27$ | $54.39 \pm 0.17$ | $35.94 \pm 0.13$ | $20.83 \pm 0.19$ |
| $\alpha=8, r_0=\frac{16}{255}, \lambda=1000$ | $86.36 \pm 0.32$ | $79.84 \pm 0.25$ | $\mathbf{72.29} \pm 0.29$ | $53.93 \pm 0.14$ | $35.06 \pm 0.22$ | $20.08 \pm 0.11$ |
| $\alpha=8, r_0=\frac{16}{255}, \lambda=2000$ | $86.10 \pm 0.15$ | $79.33 \pm 0.22$ | $72.04 \pm 0.32$ | $54.38 \pm 0.19$ | $35.36 \pm 0.27$ | $20.68 \pm 0.14$ |
| $\alpha=8, r_0=\frac{16}{255}, \lambda=3000$ | $86.05 \pm 0.27$ | $\mathbf{79.64} \pm 0.25$ | $72.24 \pm 0.28$ | $54.24 \pm 0.17$ | $35.65 \pm 0.21$ | $20.52 \pm 0.18$ |
| $\alpha=0, r_0=\frac{20}{255}, \lambda=500$ | $83.64 \pm 0.24$ | $77.28 \pm 0.31$ | $70.11 \pm 0.21$ | $54.21 \pm 0.26$ | $37.75 \pm 0.21$ | $23.85 \pm 0.20$ |
| $\alpha=0, r_0=\frac{20}{255}, \lambda=1000$ | $82.23 \pm 0.12$ | $76.20 \pm 0.24$ | $69.48 \pm 0.26$ | $\mathbf{54.82} \pm 0.25$ | $\mathbf{38.47} \pm 0.21$ | $24.80 \pm 0.20$ |
| $\alpha=3, r_0=\frac{20}{255}, \lambda=400$ | $84.51 \pm 0.08$ | $78.26 \pm 0.14$ | $70.65 \pm 0.18$ | $54.24 \pm 0.15$ | $37.28 \pm 0.15$ | $22.95 \pm 0.18$ |
| $\alpha=3, r_0=\frac{20}{255}, \lambda=800$ | $83.56 \pm 0.20$ | $77.44 \pm 0.24$ | $70.25 \pm 0.28$ | $54.22 \pm 0.23$ | $37.85 \pm 0.18$ | $23.88 \pm 0.17$ |
| $\alpha=5, r_0=\frac{20}{255}, \lambda=1000$ | $83.94 \pm 0.12$ | $77.79 \pm 0.34$ | $70.77 \pm 0.28$ | $54.39 \pm 0.27$ | $37.59 \pm 0.19$ | $23.61 \pm 0.23$ |
| $\alpha=5, r_0=\frac{32}{255}, \lambda=500$ | $81.52 \pm 0.31$ | $75.58 \pm 0.21$ | $68.03 \pm 0.24$ | $53.50 \pm 0.17$ | $38.41 \pm 0.17$ | $\mathbf{26.30} \pm 0.14$ |

**Table 7:** Clean and robust accuracy on CIFAR-10 under AA with different perturbation bounds on WRN-28-10 (with its original Batch Normalization layer). The results on different sets of hyperparameters for DyART starts from the seventh row.

| Defense | Clean | $\epsilon=\frac{2}{255}$ | $\epsilon=\frac{4}{255}$ | $\epsilon=\frac{8}{255}$ | $\epsilon=\frac{12}{255}$ | $\epsilon=\frac{16}{255}$ |
|---|---|---|---|---|---|---|
| AT | $48.09 \pm 0.38$ | $38.82 \pm 0.26$ | $30.18 \pm 0.27$ | $16.46 \pm 0.19$ | $7.74 \pm 0.20$ | $3.05 \pm 0.17$ |
| TRADES | $46.68 \pm 0.30$ | $37.84 \pm 0.21$ | $29.85 \pm 0.19$ | $16.76 \pm 0.17$ | $8.97 \pm 0.23$ | $4.43 \pm 0.11$ |
| MART | $45.51 \pm 0.29$ | $36.68 \pm 0.34$ | $29.15 \pm 0.25$ | $17.79 \pm 0.15$ | $9.91 \pm 0.17$ | $5.31 \pm 0.17$ |
| $\alpha=10, r_0=\frac{32}{255}, \lambda=500$ | $48.98 \pm 0.33$ | $38.38 \pm 0.26$ | $29.76 \pm 0.21$ | $17.30 \pm 0.23$ | $9.87 \pm 0.18$ | $5.19 \pm 0.12$ |
| $\alpha=0, r_0=\frac{16}{255}, \lambda=1000$ | $46.49 \pm 0.25$ | $37.60 \pm 0.24$ | $29.17 \pm 0.16$ | $17.0 \pm 0.19$ | $9.02 \pm 0.15$ | $4.59 \pm 0.14$ |
| $\alpha=0, r_0=\frac{20}{255}, \lambda=500$ | $49.17 \pm 0.21$ | $39.53 \pm 0.24$ | $30.20 \pm 0.22$ | $17.15 \pm 0.27$ | $9.08 \pm 0.10$ | $4.87 \pm 0.07$ |
| $\alpha=0, r_0=\frac{20}{255}, \lambda=1000$ | $43.9 \pm 0.19$ | $36.23 \pm 0.23$ | $28.89 \pm 0.20$ | $17.47 \pm 0.19$ | $10.01 \pm 0.19$ | $5.53 \pm 0.11$ |
| $\alpha=0, r_0=\frac{20}{255}, \lambda=2000$ | $42.21 \pm 0.28$ | $34.79 \pm 0.19$ | $27.66 \pm 0.21$ | $16.55 \pm 0.34$ | $9.43 \pm 0.19$ | $5.01 \pm 0.16$ |
| $\alpha=0, r_0=\frac{24}{255}, \lambda=500$ | $48.27 \pm 0.25$ | $38.75 \pm 0.20$ | $30.02 \pm 0.16$ | $18.00 \pm 0.18$ | $10.15 \pm 0.18$ | $5.56 \pm 0.08$ |
| $\alpha=5, r_0=\frac{24}{255}, \lambda=500$ | $\mathbf{50.86} \pm 0.28$ | $\mathbf{39.81} \pm 0.18$ | $30.63 \pm 0.19$ | $17.20 \pm 0.25$ | $9.36 \pm 0.11$ | $4.99 \pm 0.13$ |
| $\alpha=5, r_0=\frac{24}{255}, \lambda=1000$ | $44.59 \pm 0.32$ | $36.3 \pm 0.30$ | $28.6 \pm 0.27$ | $17.28 \pm 0.16$ | $10.16 \pm 0.17$ | $5.76 \pm 0.06$ |
| $\alpha=0, r_0=\frac{32}{255}, \lambda=500$ | $46.19 \pm 0.22$ | $37.64 \pm 0.30$ | $29.49 \pm 0.21$ | $\mathbf{18.05} \pm 0.15$ | $\mathbf{10.66} \pm 0.13$ | $\mathbf{6.27} \pm 0.09$ |
| $\alpha=3, r_0=\frac{32}{255}, \lambda=500$ | $48.56 \pm 0.20$ | $39.32 \pm 0.23$ | $30.22 \pm 0.19$ | $17.93 \pm 0.15$ | $10.22 \pm 0.15$ | $5.84 \pm 0.13$ |
| $\alpha=5, r_0=\frac{32}{255}, \lambda=500$ | $49.71 \pm 0.18$ | $39.30 \pm 0.14$ | $\mathbf{30.69} \pm 0.21$ | $18.02 \pm 0.18$ | $10.08 \pm 0.09$ | $5.65 \pm 0.12$ |
| $\alpha=5, r_0=\frac{32}{255}, \lambda=800$ | $45.90 \pm 0.34$ | $37.68 \pm 0.25$ | $29.53 \pm 0.27$ | $17.96 \pm 0.20$ | $10.55 \pm 0.15$ | $6.07 \pm 0.14$ |

**Table 8:** Clean and robust accuracy on Tiny-ImageNet under AA with different perturbation bounds on ResNet-18 (with its original Batch Normalization layer). The results on different sets of hyperparameters for DyART starts from the fourth row.

## D.3 EXPERIMENTAL RESULTS WITH ADDITIONAL DATA

Recent work shows that generative models which are trained solely on the original training data can be used to drastically improve the adversarial robustness performance (Rebuffi et al., 2021; Gowal et al., 2021). In this section, we demonstrate the results of DyART on CIFAR-10 using 1M additional data from DDPM (Ho et al., 2020). The experimental setting follows previous works (Rebuffi et al., 2021; Gowal et al., 2020) and their PyTorch implementation [9], which is consistent with Section D.1 except that now we run 600 epochs with batch size 512 using the additional data. The initial learning rate is still set to 0.1. Note that in the original implementation in previous works (Rebuffi et al., 2021; Gowal et al., 2020), 800 epochs are run with batch size 1024, with initial learning rate 0.4, which could improve the performance compared with 600 epochs and 512 batch size that we use for DyART. All experiments in this section use the original batch normalization layer in WRN.

**Hyperparameters DyART with additional data**   We use WRN-28-10 and we choose $\alpha = 3, r_0 = \frac{16}{255}, \lambda = 800, \beta = 5$ and apply gradient clipping with threshold 0.1. We use 10 epochs of natural training as the burn-in period.

**Performance of DyART with additional data**   The experimental results are shown in Table 9. We have the following two observations: (1) Compared with the results in Table 7, it is clear that the additional data can drastically improve the robust accuracy of DyART (about 6% boost in robust accuracy under $\epsilon = \frac{8}{255}$). (2) Compared with the state-of-the-art results in Rebuffi et al. (2021), DyART achieves higher clean accuracy and comparable robust accuracy under $\epsilon = \frac{8}{255}$ (Rebuffi et al. (2021) does not report robust accuracy under other perturbation bounds.).

| Defense | Architecture | Clean | $\epsilon = \frac{2}{255}$ | $\epsilon = \frac{4}{255}$ | $\epsilon = \frac{8}{255}$ | $\epsilon = \frac{12}{255}$ | $\epsilon = \frac{16}{255}$ |
|---|---|---|---|---|---|---|---|
| Rebuffi et al. (2021) | WRN-28-10 | 85.97 | NA | NA | 60.73 | NA | NA |
| Rebuffi et al. (2021) | WRN-70-16 | 86.94 | NA | NA | 63.58 | NA | NA |
| DyART | WRN-28-10 | 87.46 | 82.30 | 75.68 | 60.35 | 42.38 | 26.42 |

**Table 9:** Clean and robust accuracy on CIFAR-10 under $l_\infty$ AutoAttack with different perturbation sizes when 1M additional generated data from DDPM is used for training.

## D.4 FURTHER ANALYSIS ON DECISION BOUNDARY DYNAMICS

In Section 4.2 and Section 6.2 we have presented the dynamics of both AT and DyART on the same pretrained models using the same batch of data at one iteration. In this section, we demonstrate the dynamics of the decision boundary throughout the whole training process.

**Experiment setting.**   To study the decision boundary dynamics throughout the training process, we train a ResNet-18 (He et al., 2016a) with group normalization (GN) (Wu & He, 2018) on CIFAR-10 using (1) Adversarial Training with 10-step PGD under $\ell_\infty$ perturbation with $\epsilon = \frac{8}{255}$ from scratch; (2) DyART with $\alpha = 8, \lambda = 400, r_0 = \frac{16}{255}$ from a naturally pretrained model. The models are trained with a initial learning rate of 0.01 and the learning rate decays to 0.001 at 20000 iteration. At each iteration, we compute the proportion of negative speed among points with margins smaller than $\frac{8}{255}$ that are regarded as vulnerable.

**Conflicting dynamics throughout training**   In Figure 6, the clean and robust accuracy of both methods are shown, along with the proportion of negative speed among vulnerable points. We apply curve smoothing for negative speed proportion plot for better visualization. Note that we omit the initial part of training (first 10 epochs) since at this initial stage, there are not enough correctly classified data points but speed and margin are only defined for these points. We can see that both methods exhibit some degree of robust overfitting, where the training robust accuracy becomes larger than the test robust accuracy. In addition, the conflicting dynamics exists throughout the whole training process, since the proportion of negative speed is never zero. We can see that DyART consistently has less conflicting dynamics than AT. Interestingly, the proportion of negative speed decreases over time during training for both methods. The connection between the decreasing degree

---

[9]UncoveringATLimits's Github

of conflicting dynamics on the training data and the robust overfitting phenomenon is left for future research.

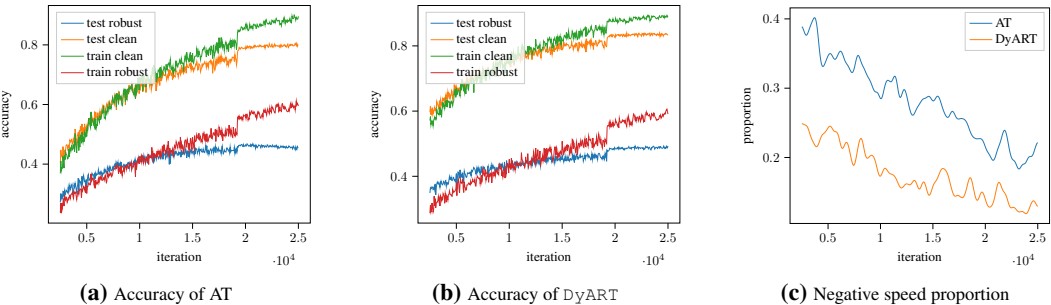

**(a)** Accuracy of AT      **(b)** Accuracy of `DyART`      **(c)** Negative speed proportion

**Figure 6:** The accuracy of AT and `DyART` as well as the proportion of negative speed among points whose margins are smaller than $\frac{8}{255}$.

## D.5 RUN TIME ANALYSIS

In this section we provide the run time analysis. The main computational bottleneck for `DyART` is finding the closest boundary points, which is an iterative algorithm adapted from FAB. Each iteration costs one back-propagation, which is the same as Projected Gradient Descent (PGD). Once we find these closest boundary point candidates, we check if the KKT condition is approximately satisfied and filter out points that do not meet the KKT condition. The computational cost of this step takes one back-propagation.

We use the torch.cuda.Event functionality in PyTorch to measure the execution time for one iteration of each method. In the case of `DyART`, this means measuring the total time of finding the closest boundary points and do back propagation using the full loss function. We use ResNet-18 with GroupNorm on a batch size of 128 on the CIFAR10 dataset. We use one NVDIA RTX A4000. The results are as follows:

- Natural training: $46 \pm 0.9$ ms
- AT (PGD-10 on Cross-Entropy loss): $531 \pm 5.2$ ms
- TRADES (PGD-10 on KL divergence loss): $573 \pm 2.8$ ms
- DyART (10 steps for finding the closest boundary point): $743 \pm 10.3$ ms
- DyART (20 steps for finding the closest boundary point): $1171 \pm 17.8$ ms

Developing faster algorithms for finding the closest boundary points is left for future research.

