# OpenReview forum: "Exploring and Exploiting Decision Boundary Dynamics for Adversarial Robustness"
_ICLR.cc/2023/Conference — ICLR 2023 poster_

### Official Review · Reviewer_wA1W · 2022-10-17

**Confidence:** 4
**Correctness:** 3
**Technical Novelty And Significance:** 3
**Empirical Novelty And Significance:** 3
**Recommendation:** 6

**Clarity, Quality, Novelty And Reproducibility:**

The writing of this paper is good and easy to follow, but some places may need further clarification. The study on decision boundary dynamics is novel and interesting. However, for the adversarial training part, there is a lack of comparison or discussion on some recent adversarial training papers.

**Strength And Weaknesses:**

Strength:

+ This paper shows that there are conflicting dynamics in adversarial training. It is interesting to know that the decision boundary of some examples becomes even worse after training.
+ This paper proposes a novel adversarial training method, DyART, which has a novel loss to alleviate the conflicting dynamics.
+ By evaluating on CIFAR10 and Tiny ImageNet, the proposed method outperforms other methods.
+ The paper is well-written and easy to follow.

Weakness & Questions:

- In Section 4.1, this paper studies how boundary changes by evaluating CIFAR10 models. The decision boundary speed is evaluated by training an additional iteration on pretrained models. For this experiment, I am not quite clear about some settings:

1. Why do you use a pretrained model to calculate the dynamics? Why don’t train a model from scratch and observe the dynamics?
2. The dynamics are only calculated for one batch and one iteration, which makes it more like a case study or qualitative analysis. Why don’t you collect the dynamic across the whole training process and analyze the speed at different epochs? Maybe this conflict only happens at the beginning of the training and will be stable later.
3. I don’t get the reason behind replacing BN with GN, which may indicate that the proposed method to evaluate the decision boundary speed is not general enough.

- In Section 6.1, I notice that a validation set is used to mitigate the robust overfitting. How often do you calculate the validation accuracy, and do you apply the same thing on other baseline methods to mitigate robust overfitting?

- Although the proposed method outperforms selected baseline methods, there are other state-of-the-art (SOTA) methods reported to have better performance than selected baseline methods in https://robustbench.github.io/. What is the advantage of the proposed method compared to other more recent SOTA adversarial training methods?

**Summary Of The Paper:**

This paper first proposes a continuous-time framework for quantifying the speed of decision boundary changes and empirically shows conflicting dynamics in adversarial training: the decision boundary moves closer to some examples even if the model just learns these examples. This paper proposes a novel adversarial training method, DyART, aiming to increase margins to alleviate the conflicting dynamics. By evaluating DyART on CIFAR10 and Tiny-Imagenet datasets, this paper shows that the proposed method performs better.

**Summary Of The Review:**

I thoroughly enjoy reading this paper, but I think that there needs more clarification. One concern is comparisons to some recent related works on adversarial training are missing.

---

> ### Author Response · Authors · 2022-11-14
> **Clarification on experiments**
>
>
> > Q4: In Section 6.1, I notice that a validation set is used to mitigate the robust overfitting. How often do you calculate the validation accuracy, and do you apply the same thing on other baseline methods to mitigate robust overfitting?
>
> In all experiments (our method and baselines), we compute the clean and robust accuracy at the end of each epoch on the validation set. The robust accuracy is computed using PGD-20 using margin loss. For all methods, we choose the model with the highest robust accuracy on the validation set, and perform the Auto-Attack on the test set. We have updated the paper (Appendix C.1) to further explain this detail. Thank you for this question!
>
> ---
> > Q5: Although the proposed method outperforms selected baseline methods, there are other state-of-the-art (SOTA) methods reported to have better performance than selected baseline methods in https://robustbench.github.io/. What is the advantage of the proposed method compared to other more recent SOTA adversarial training methods?
>
>
>
> On the benchmark page, the most significant factor for boosting the robustness performance is the usage of more data (either additionally collected data or synthetic data from generative models[1]). While this line of works focuses on how to utilize more data (and they still use TRADES[2] as the base robust training method) and achieves impressive improvement, our work is orthogonal to them: we focuses on designing a robust training method that directly increase margins and prioritize increasing smaller ones. Our method could be combined with these data-based methods to improve performance, but this is out of scope of this paper. The main focus and contribution of this paper is to understand the dynamics of the decision boundary in a more rigorous way and exploit the dynamics to improve robust training.
>
> We added this line of work and compare them with our method in Appendix A.
>
> ---
>
> Thank you again for your time and effort in reviewing our paper! Please let us know if the above explanations do not address your concerns. We are happy to answer any further questions.

---

> ### Author Response · Authors · 2022-11-14
> **Clarification on normalization layers**
>
> > Q3: I don’t get the reason behind replacing BN with GN, which may indicate that the proposed method to evaluate the decision boundary speed is not general enough.
>
> To explain why we need to replace BN with GN (or any other normalization layers that do not result in prediction inconsistency during training and test time), we first recall that the speed is computed as $s(x\_{i},t) = \frac{1}{\|\nabla\_{x} \phi^{y\_{i}}(\hat{x}\_{i},\theta(t))\|\_{q}} \nabla\_{\theta}\phi^{y\_{i}}(\hat{x}\_{i},\theta(t))\cdot \theta'(t)$ (Proposition 3).
>
> At each iteration, when BN is used, the (BN layer) parameters of neural network $\theta\_{\text{train}}$ using training mode and $\theta\_{\text{eval}}$ using evaluation mode are different. When finding $\hat{x}\_{i}$, we need to freeze the parameters of the network so that the decision boundary does not change, and thus $\theta\_{\text{eval}}$ is used. However, $\theta'(t)$ is computed based on $\theta\_{\text{train}}$ using the training mode. Because $\theta\_{\text{train}} \neq \theta\_{\text{eval}}$ when BN is used, it is not reasonable to compute the speed using the $\theta'(t)$ and the found $\hat{x}\_{i}$. On the other hand, when GN is used, $\theta\_{\text{train}} = \theta\_{\text{eval}}$ and thus the speed can be computed.
>
> Our work focuses on a framework to track the dynamics of the decision boundary and is general enough to handle all kinds of training algorithms (in addition to robust training algorithms) and most architectures. In the case of BN, due to its inconsistency between training and evaluation, the training dynamics of the decision boundary is not well-defined. However, this does not reduce the significance of our framework: rigorously and directly exploring and exploiting decision boundary dynamics for robust training. To extend our framework to BN, one potential solution is to mathematically derive how BN parameters change in the training mode affects the change of BN parameters in the test mode. This extension is left for future.
>
>
> In addition, we would like to mention that more and more architectures such as transformers for vision [1] and NLP [2] are replacing BN with other normalization such as Group Normalization and Layer Normalization, which can be studied using our framework. For convolutional networks, the current state of the art architecture[3] does not use BN anymore. We believe that our framework is general and is important for future research.
>
> Ref:
> [1] Dosovitskiy, Alexey, et al. "An image is worth 16x16 words: Transformers for image recognition at scale." ICLR 2021.
> [2] Devlin, Jacob, et al. "Bert: Pre-training of deep bidirectional transformers for language understanding." arXiv preprint arXiv:1810.04805 (2018)
> [3] Liu, Zhuang, et al. "A convnet for the 2020s." CVPR 2022.

---

> ### Author Response · Authors · 2022-11-14
> **Clarification on decision boundary dynamics during training**
>
> We thank Reviewer wA1W for the detailed and insightful feedback. We are encouraged that Reviewer wA1W finds our framework novel and enjoys reading our paper. Below we address Reviewer wA1W's concerns in detail, and provide additional experimental results.
>
> ---
>
> >  Q1: Why do you use a pretrained model to calculate the dynamics? Why don’t train a model from scratch and observe the dynamics?
>
> Since margins and speed are only defined for correctly classified points, it is more reasonable to measure the dynamics of the decision boundary when the neural network has a relatively high clean accuracy. In this case, a large proportion of data points will have their well-defined margins and speed. This is why we first train the model to have a descent clean accuracy, and then observe its dynamics, instead of observing the dynamics from the beginning.
>
> ---
>
> > Q2: The dynamics are only calculated for one batch and one iteration, which makes it more like a case study or qualitative analysis. Why don’t you collect the dynamic across the whole training process and analyze the speed at different epochs? Maybe this conflict only happens at the beginning of the training and will be stable later.
>
> Thank you for this important question! In the paper (Section 4.2 and 6.2), we do consider the decision boundary dynamics in different stages of the training process by considering (1) a partially trained model using natural training, which stands for the start of adversarial training; (2) a partially trained model using adversarial training, which stands for the middle of adversarial training process. In Figure 3 we can see that the margins get larger from (1) to (2) (due to adversarial training), but both cases have a large proportion of negative speed, suggesting that the conflicting dynamics exists during the whole adversarial training process.
>
> To further analyze the decision boundary dynamics throughout the whole training process, in Appendix D.3, we added a plot of the negative speed proportion among points with margins smaller than $\frac{8}{255}$ (i.e. vulnerable points). We make the following observations
>
> 1. Conflicting dynamics exists throughout training: the proportion of negative speed never goes to zero.
> 2. Our proposed DyART consistently has lower negative speed proportion compared with AT, throughout the whole training process.
> 3. For both DyART and AT, the negative speed proportion decreases over time (but is always non-zero). This might be related to robust overfitting phenomenon since both methods shows some degree of overfitting on the training data. Understanding the connection between such decrease in conflicting dynamics and robust overfitting is left for future research.

---

> ### Author Response · Authors · 2022-11-27
> **Does our response address your concerns?**
>
> Dear reviewer wA1W,
>
> As the first stage of the review discussion has ended, we would like to kindly ask you to review our revised paper as well as our response and consider making adjustments to the scores. Please let us know if there are any other questions. We would appreciate the opportunity to engage further if needed.
>
> Best regards,
>
> Paper3155 Authors

---

### Official Review · Reviewer_o95F · 2022-10-23

**Confidence:** 3
**Clarity, Quality, Novelty And Reproducibility:** Code is not provided.
**Correctness:** 3
**Technical Novelty And Significance:** 3
**Empirical Novelty And Significance:** Not applicable
**Recommendation:** 6

**Strength And Weaknesses:**

Strength:
(1) The paper give sound theoretical analysis.
(2) The insight of giving high priority to examples with smaller margins is interesting.

Weakness:
(1) This paper identified the phenomenon that during adversarial training, the decision boundary moves away from some vulnerable points but simultaneously moves closer to others, decreasing their margins. However, it is well-known that adversarial training suffers from the overfitting issue, i.e., adversarial examples in training can be well classified in the late training stage. So, does the identified phenomenon only exists in the early adversarial training stage?
(2) Experimental setting is not consistent with previous work, e.g., TRADES uses wideresnet34-10 and achieves 53.08% while wideresnet28-10 is adopted and only reports 49.3% in this paper.
(3) Comparisons with state-of-the-art methods are missed, like AWP[1], and LBGAT[2].
(4)  Experiments on CIFAR-100 should be included to verify consistent improvements of the proposed method.

[1] Adversarial Weight Perturbation Helps Robust Generalization. NeurIPS 2020.
[2] Learnable Boundary Guided Adversarial Training. ICCV 2021.



**Summary Of The Paper:**

This paper aims to improve the adversarial robustness of deep models.
I. The authors observe that during traditional adversarial training, margins to the decision boundary for some examples are enlarged while margins to the decision boundary for other examples even become smaller.
II. Based on this phenomenon, the paper proposes dynamic-aware robust training,  giving high priority to examples suffering from smaller margins.
III. Experiments on CIFAR-10 and Tiny-ImageNet show some improvements over baselines.

**Summary Of The Review:**

This paper aims to improve model adversarial robustness with a dynamic-aware robust training strategy. However, the inconsistent experimental setting and lack of comparisons with state-of-the-art methods make it unconvinced.

---

> ### Author Response · Authors · 2022-11-14
> **Clarification on decision boundary dynamics and experiments on CIFAR-100**
>
> We thank Reviewer o95F for the detailed and insightful feedback. We are encouraged that Reviewer o95F finds our paper sound and interesting. Below we address Reviewer o95F's concerns in detail, and provide additional experimental results.
>
> ---
>
> >  Q1: This paper identified the phenomenon that during adversarial training, the decision boundary moves away from some vulnerable points but simultaneously moves closer to others, decreasing their margins. However, it is well-known that adversarial training suffers from the overfitting issue, i.e., adversarial examples in training can be well classified in the late training stage. So, does the identified phenomenon only exists in the early adversarial training stage?
>
> Thank you for this important question! The conflicting dynamics exists throughout the training process, including both early and final stages. In the experiment in Section 4.2, we show the decision boundary dynamics for a partially robust model (trained via adversarial training) with 42% robust accuracy on the test set and 53% robust accuracy on the training set (so the robust over-fitting occurs). We can see in the second figure in Figure 3 that there are a lot of negative speed values with margins smaller than $\frac{8}{255}$, indicating that the conflicting dynamics exists even when robust over-fitting occurs.
>
> To gain more insights, in Appendix D.3, we added a plot of the negative speed proportion among points with margins smaller than $\frac{8}{255}$ (i.e. vulnerable points). We make the following observations
>
> 1. Conflicting dynamics exists throughout training: the proportion of negative speed never goes to zero.
> 2. Our proposed DyART consistently has lower negative speed proportion compared with AT, throughout the whole training process.
> 3. For both DyART and AT, the negative speed proportion decreases over time (but is always non-zero). This might be related to robust overfitting phenomenon since both methods shows some degree of overfitting on the training data. Understanding the connection between such decrease in conflicting dynamics and robust overfitting is left for future research.
>
> ---
>
> > Q2: Experimental setting is not consistent with previous work, e.g., TRADES uses wideresnet34-10 and achieves 53.08% while wideresnet28-10 is adopted and only reports 49.3% in this paper.
>
> WideResnet-28-10 is  commonly used in the robust training literature. To quote in one recent paper [1], "We note that most work on adversarial robustness on Cifar-10 use either a WRN-34-10 or a WRN-28-10 network, with WRN-34-20 being another popular option." For all the baselines, we follow the correct hyper-parameters suggested by the original papers, the standard evaluation and training protocol.
>
> Ref:
> [1] Gowal, Sven, et al. "Uncovering the limits of adversarial training against norm-bounded adversarial examples." arXiv preprint arXiv:2010.03593 (2020).
>
> ---
>
> > Q3 Comparisons with state-of-the-art methods are missed, like AWP[1], and LBGAT[2].
>
>
> Thank you for pointing out these relevant literature! We have updated our literature review part in Section 2 and in Appendix A.
>
> Ref:
> [1] Adversarial Weight Perturbation Helps Robust Generalization. NeurIPS 2020.
> [2] Learnable Boundary Guided Adversarial Training. ICCV 2021.
>
> ---
>
> > Q4: Experiments on CIFAR-100 should be included to verify consistent improvements of the proposed method.
>
> In this paper, we use two datasets: CIFAR-10 and Tiny-ImageNet. CIFAR-10 has fewer classes (10 classes) and Tiny-ImageNet has more classes (200 classes). Note that Tiny-ImageNet is more challenging than CIFAR-100. Our proposed method achieves better performance than baselines on both CIFAR-10 and Tiny-ImageNet, which is sufficient to demonstrate its effectiveness.
>
> We did a preliminary experiment of CIFAR-100. The hyperparamters of DyART is $\alpha=3, \lambda=800, r_1 = 16$. All experiments are ran for 100 epochs, and the robust accuracy is measured on the validation set of size 1024 at each epoch using PGD-20. The model with the highest validation robust accuracy is chosen for each method, and we use AutoAttack to measure its  robustness performance.
>
> | Defense | Clean | $\epsilon=4/255$ | $\epsilon=8/255$ | $\epsilon=12/255$ |
> |:-------- |:--------:| --------:|--------:|--------:|
> | AT     |  54.64   |     37.75 |     24.99 |     15.23 |
> | TRADES     |  58.65   |     39.70 |    25.68 |    15.74 |
> | MART     |   48.80   |     35.26 |    24.98 |    15.72 |
> | Ours     |   59.72   |     41.09 |    26.02 |    15.26 |
>
> We can see that our method has higher clean accuracy and higher robust accuracy under $\epsilon = \frac{4}{255}, \frac{8}{255}$. This shows that our method is also effective on the CIFAR-100 dataset.

---

> > ### Comment · Reviewer_o95F · 2022-11-19
> > **Respondse to authors**
> >
> > (1) "Partially trained model".
> > I guess the "partially trained model" here represents that the model is not fully trained by adversarial training. In this case, the model may not overfit to adversarial examples in training. So It only achieves 53% robust accuracy on the training set.
> >
> > (2) Experimental setting is not consistent with previous work, e.g., TRADES uses wideresnet34-10 and achieves 53.08% while wideresnet28-10 is adopted and only reports 49.3% in this paper.
> >
> > Though the authors claim that default hyper-parameters are adopted in their experiments, the large performance gap between wideresnet34-10 and wideresnet28-10 (53.08% vs 49.3%) is surprising.
> >
> > (3) Comparisons with state-of-the-art methods are missed, like AWP[1], and LBGAT[2].
> >
> > Training with large amounts of additional data is time-consuming. However,
> >
> > AWP and LBGAT are trained without additional data. The paper should compare these kinds of methods.
> >
> > (4) Experimental results on CIFAR-100.
> >      AWP and LBGAT achieve 28.86% and 29.33% robust accuracy with wideresnet34-10 respectively, largely outperforming the proposed method (26.02).

---

> > > ### Author Response · Authors · 2022-11-19
> > > **Further Clarification**
> > >
> > > Thank you for your follow-up comments!
> > >
> > > > Q1: "Partially trained model". I guess the "partially trained model" here represents that the model is not fully trained by adversarial training. In this case, the model may not overfit to adversarial examples in training. So It only achieves 53% robust accuracy on the training set.
> > >
> > > In Appendix D.3, we also provide the decision boundary dynamics throughout the **whole** training process, including the robust-overfitting phase (when the learning rate decays and the robust accuracy gap between training and test set gets very large). We make the following observations:
> > >
> > > 1. Conflicting dynamics exists throughout training: the proportion of negative speed never goes to zero.
> > > 2. Our proposed DyART consistently has lower negative speed proportion compared with AT, throughout the whole training process.
> > > 3. For both DyART and AT, the negative speed proportion decreases over time (but is always non-zero). This might be related to robust overfitting phenomenon since both methods shows some degree of overfitting on the training data. Understanding the connection between such decrease in conflicting dynamics and robust overfitting is left for future research.
> > >
> > > ---
> > >
> > > > Q2: Experimental setting is not consistent with previous work, e.g., TRADES uses wideresnet34-10 and achieves 53.08% while wideresnet28-10 is adopted and only reports 49.3% in this paper. Though the authors claim that default hyper-parameters are adopted in their experiments, the large performance gap between wideresnet34-10 and wideresnet28-10 (53.08% vs 49.3%) is surprising.
> > >
> > > The performance gap comes from two parts:
> > >
> > > 1. model size (we use a smaller network: WRN-28-10)
> > > 2. In our experiments all Batch Normalization layers are replaced with Group Normalization layers.
> > >
> > > However, we would like to stress that using the same architectures, our method outperforms the baselines. We will work on adapting our method to Batch Normalization in the future.
> > >
> > > ---
> > >
> > > > Q3 & Q4: Comparisons with state-of-the-art methods are missed, like AWP[1], and LBGAT[2]. Training with large amounts of additional data is time-consuming. However, AWP and LBGAT are trained without additional data. The paper should compare these kinds of methods.
> > > Experimental results on CIFAR-100. AWP and LBGAT achieve 28.86% and 29.33% robust accuracy with wideresnet34-10 respectively, largely outperforming the proposed method (26.02).
> > >
> > > Thank you for pointing out these baselines! Due to the limited time, we leave the performance comparison for the future. We remark on two reasons for the performance gap:
> > >
> > > 1. **[Model Architecture]** As explained in Q2, the architecture size and normalization layers are different
> > > 2. **[Data splitting]** We check the code base of both methods and it seems that they do model selection directly on the test set, while we select models using a validation set of size 1024. Since in their implementations, the test set is already used before evaluation, it is reasonable that their performance numbers on the test set are higher.
> > >
> > > ---
> > >
> > > Thank you again for your follow-up comments!

---

> > > ### Author Response · Authors · 2022-12-06
> > > **Our method now works with BN and achieves SOTA under the BN setting**
> > >
> > > Dear reviewer,
> > >
> > > In previous experiments we replaced the Batch Normalization (BN) with Group Normalization (GN) so that the decision boundary during training and test time are the same. We are aware that directly replacing BN may decrease the performance of the network and not being able to deal with BN is one limitation of our method. Although we would like to mention that under GN settings, we did achieve state-of-the-art performance.
> > >
> > > After further experiments, We find that when combined with **gradient clipping**, our method works for BN. That is, at each iteration we clip the gradient when the gradient norm of the loss function is larger than some pre-defined threshold. We find that this helps stablizing the training of BN parameters.
> > >
> > > Below we present the experimental results (under AutoAttack) using WRN-28-10 (without any modification to the architecture). We also compare our method with AWP and LBGAT as requested. We choose to combine AWP and LBGAT with TRADES since they have higher robust accuracy. In the compared baselines, our method is only 0.1% lower in robust accuracy than AWP, but is 1.15% higher in clean accuracy and has higher robust accuracy than other baselines. We would like to mention that our method goes out of the min-max framework of traditional methods and exploit the decision boundary dynamics, and still achieves the state-of-the-art performance.
> > >
> > > | Defense | Clean | $\epsilon=8/255$ |
> > > |:-------- |:--------:| --------:|
> > > | TRADES     |  83.38   |     51.94 |
> > > | AT     |  85.06   |     50.57 |
> > > | LBGAT + TRADES ($\alpha=6$)     |  81.26   |  52.45 |
> > > | AWP + TRADES     |  84.05   |     54.16 |
> > > | Ours     |   85.29   |     54.05  |
> > >
> > > We will include our experimental results with BN and additional baselines in the camera ready version of the paper.
> > >
> > > As we have solved the concern about the performance, we would like to kindly ask you consider making adjustments to the scores. Please let us know if there are any other questions. We would appreciate the opportunity to engage further if needed.
> > >
> > > Best regards,
> > >
> > > Paper3155 Authors
> > >
> > > ---
> > > Ref:
> > > [1] Adversarial Weight Perturbation Helps Robust Generalization. NeurIPS 2020.
> > > [2] Learnable Boundary Guided Adversarial Training. ICCV 2021.

---

> > > > ### Comment · Reviewer_o95F · 2022-12-07
> > > > **Response to the authors**
> > > >
> > > > Thanks for the reply from the authors.
> > > >
> > > > The authors provide experimental comparisons with recent state-of-the-art methods, like AWP on CIFAR-10. The reported numbers seem inconsistent with that in their original paper (AWP original paper vs. reported here 56.17% vs 54.16%), The reason behind this can be different experimental settings.
> > > >
> > > > Considering their theoretical contributions, I would like to raise my score from 5 to 6.

---

> ### Author Response · Authors · 2022-11-14
> **Our novelty and contributions**
>
> We would like to emphasize two major contributions of our work:
>
> (a) [**Principled framework for the decision boundary dynamics**]
> To our best knowledge, our work is the first to explicitly formulate and rigorously derive the closed-form expression for the dynamics of the decision boundary during training: we define the speed of the decision boundary as the rate of change of the margin, and derive a closed-form expression for the speed that can be computed in practice. The question of how decision boundary changes is crucial in deep learning, and our framework is useful for understanding the training process of deep learning and will inspire future research.
>
> [**Useful insights**] Our framework reveals the conflicting dynamics of decision boundary in adversarial training, that it neither converges (zero speed), nor always goes away from data points. Instead, it oscillates w.r.t. data points. We believe that such **insight** is very helpful for robust training community.
>
> (b) [**Explicitly optimize margins and control dynamics**]
> By deriving a closed-form expression for the gradient of margin, our work make it possible to directly optimize margins using gradient-based optimization methods. For robust training, we design a loss function that directly relates to margins and prioritizes increasing smaller margins. Note that although increasing margins is a reasonable goal for robust training, previous robust training methods **cannot** directly increase margins because it was previously unknown how to compute the gradient of margins. Our proposed method is the first to directly manipulate margins and achieves competitive robustness performance.
>
> ---
>
> Thank you again for your time and effort in reviewing our paper! Please let us know if the above explanations do not address your concerns. We are happy to answer any further questions.

---

### Official Review · Reviewer_1WcD · 2022-10-24

**Confidence:** 5
**Correctness:** 4
**Technical Novelty And Significance:** 4
**Empirical Novelty And Significance:** 4
**Recommendation:** 10

**Clarity, Quality, Novelty And Reproducibility:**

- **Clarity**: The paper is very clearly written. The proofs of the main theorems are also easy to follow.
- **Quality**: The paper has a high technical quality. The mathematical derivations are elegant, and the robustness evaluation follows the right protocols established by prior art.
- **Novelty**: As far as I know, the ideas and results in this paper are novel. A few citations could be included to a few papers that discuss similar topics.
- **Reproducibility**: The main results of this work are probably reproducible. Howevere, a few extra ablation studies would simplify the adoption of DyART as a robustness method.

**Strength And Weaknesses:**

# Strengths
1. **A fresh and effective idea**: There are thousands of papers on adversarial robustness, all of them trying to improve the robustness of neural networks in different ways. However, among the empirical defenses, only adversarial training and its variants have managed to stand the test of time. This paper tries to tackle the same problem, using a very different approach (regularizing the exact distance to the boundary), which seems to be very effective.
2. **Brilliant derivations**: The derived mathematical expressions on the movement of the decision boundary are absolutely brilliant and very noteworthy. Above all, the main value of this paper does not lie on their idea to obtain robustness, but the fact that the authors could manage to execute it by cleverly and elegantly deriving tractable expressions to track the movement of the decision boundary. The main theorems of the paper do not require advanced math to be proved, and are very intuitive. This is a very strong point.
3. **Very competitive robustness numbers and a correct evaluation**: The reported results are competitive with other methods and the authors seem to have followed the right protocols to evaluate the robustness of their models.
4. **Clean writing**: The paper is very well-written and it is easy to read. The storyline flows nicely, and the theoretical bits, including the proofs, are easy to follow.

# Weaknesses

1. **Lack of ablations on new method**: I find the paper would be stronger if, besides reporting best performances on two datasets, it also described how different hyperparameters affect the performance and dynamics of their proposed method. How do $\alpha, \lambda, r_{0}$ affect performance? And the burn-in period? Knowing the effect of these parameters and the sensitivity of the method to their exact values is fundamental for its wide adoption. It would also be nice to provide runtime numbers of DyART and compare them to the competing methods.
2. **(Minor) Missing relevant citations**: In general, I find the paper is missing citations to a few relevant works that also have studied the dynamics of the margins of adversarial training or been inspired by them to improve the performance of robust models:
   - R. Rade, S. Moosavi-Dezfooli. Reducing excessive to achieve a better accuracy vs. robustness tradeoff. ICLR 2022
   - G. Ortiz-Jimenez, A. Modas, S. Moosavi-Dezfooli, P. Frossard. Hold me tight! Influence of discriminative features on deep network boundaries. NeurIPS 2020.
   - F. Tramèr, J. B., Nicholas Carlini, N. Papernot, J. Jacobsen. Fundamental Tradeoffs between Invariance and Sensitivity to Adversarial Perturbations. ICML 2020

3. **(Minor) Figure 3 and Figure 5 could be improved**. Although informative, I find the design of this two plots a bit suboptimal. In Fig. 3, instead of showing just a snapshot of the margin-speed distributions, it would be more interesting, in my opinion to show how this distribution evolves with training. One option to do that is to show the statistic described in the caption through time (i.e., what is the proportion of points with negative speed below a given margin at step t). The problem with Fig. 5, on the other hand, is that the graph is a bit confusing. What is the meaning of the bar heights, and why is it the same for AT and DyART?


**Summary Of The Paper:**

This paper proposes DyART: a new regularization method that encourages that the decision boundaries of a neural network move away from the data points with the smallest marginss during training. This regularizer is motivated by the observation that during adversarial training, the distance to the decision boundary does not grow uniformly for all data points, and instead decreases for a large portion of the training set. To alleviate this issue, the authors derive a closed-form expression for the margin velocity around a given data point that can be tractably regularized during training. In two datasets (CIFAR10 and TinyImageNet), DyART outperdorms can achieve stronger robustnesss than SOTA methods based on adversarial training, despite being based on a conceptually very different approach.

**Summary Of The Review:**

Overall, I believe this paper clearly deserves to be accepted to ICLR. It brings forward a novel idea based on interesting insights and provides useful mathematical derivations for the community. The empirical results also seem to indicate that DyART could be a strong contender for the SOTA in adversarial robustness.

If the ablation studies mention in the **weaknesses** section of my review were included in the paper, I would be open to increase my score even further.

---

> ### Author Response · Authors · 2022-11-14
> **Clarification on figures of decision boundary dynamics**
>
>
> > Q3.1: (Minor) Figure 3 and Figure 5 could be improved. Although informative, I find the design of this two plots a bit suboptimal. In Fig. 3, instead of showing just a snapshot of the margin-speed distributions, it would be more interesting, in my opinion to show how this distribution evolves with training. One option to do that is to show the statistic described in the caption through time (i.e., what is the proportion of points with negative speed below a given margin at step t).
>
> Thank you for the suggestion! In the paper (Section 4.2 and 6.2), we do consider the decision boundary dynamics in different stages of the training process by considering (1) a partially trained model using natural training, which stands for the start of adversarial training; (2) a partially trained model using adversarial training, which stands for the middle of adversarial training process. In Figure 3 we can see that the margins get larger from (1) to (2) (due to adversarial training), but both cases have a large proportion of negative speed, suggesting that the conflicting dynamics exists during the whole adversarial training process.
>
> To further analyze the decision boundary dynamics throughout the whole training process, in Appendix D.3, we added a plot of the negative speed proportion among points with margins smaller than $\frac{8}{255}$ (i.e. vulnerable points). We make the following observations
>
> 1. Conflicting dynamics exists throughout training: the proportion of negative speed never goes to zero.
> 2. Our proposed DyART consistently has lower negative speed proportion compared with AT, throughout the whole training process.
> 3. For both DyART and AT, the negative speed proportion decreases over time (but is always non-zero). This might be related to robust overfitting phenomenon since both methods shows some degree of overfitting on the training data. Understanding the connection between such decrease in conflicting dynamics and robust overfitting is left for future research.
>
> > Q3.2 The problem with Fig. 5, on the other hand, is that the graph is a bit confusing. What is the meaning of the bar heights, and why is it the same for AT and DyART?
>
> Thank you for the clarification question! The bar height stands for the proportion of points whose margins are within the corresponding interval. Because both AT and DyART runs on the **same** model for one iteration at the **same** batch of data, their bar heights are the same. For example, the bar height for the interval $[0,\frac{4}{255}]$ is the proportion of margins inside this interval for the batch of data and model parameters that are used by both data.
>
> ---
>
> Thank you again for your time and effort in reviewing our paper! Please let us know if the above explanations do not address your concerns. We are happy to answer any further questions.

---

> > ### Comment · Reviewer_1WcD · 2022-11-15
> > **Thank you for your reply**
> >
> > Thank you very much for your clarifications and for following the recommendations of most reviewers and providing new experiments in support of your work. The new ablation studies will be very useful for the community, and I would encourage the authors to add their runtime discussion to the paper. Overall, **I clearly think this paper deserves to be accepted at the conference and I will strongly argue in favour of that**.
> >
> > The main concern of other reviewers seems to be the lack of comparisons with the strongest baselines in adversarial robustness, which as the authors argue mostly exploit additional data to further boost robustness. In this sense, I symphatize with the intuitions of the authors that DyART is improving robustness in an orthogonal way, and will *probably* further improve robustness in those settings. However, this is still a hypothesis, and should be tested to fully confirm this. In this regard, it is important to mention that these stronger baselines are not necessarily much heavier in terms of compute (roughly 4 times more than the AT setting in this work) so it would have been feasible to provide some experiments with them in this work.

---

> > > ### Author Response · Authors · 2022-11-18
> > > **Thank you for your follow-up comments**
> > >
> > > Thank you for your follow-up comments and questions!
> > >
> > > > Q1: Thank you very much for your clarifications and for following the recommendations of most reviewers and providing new experiments in support of your work. The new ablation studies will be very useful for the community, and I would encourage the authors to add their runtime discussion to the paper. Overall, I clearly think this paper deserves to be accepted at the conference and I will strongly argue in favour of that.
> > >
> > > Thank you for supporting our work! We have added the run time discussion in Appendix D.4.
> > >
> > > > Q2: The main concern of other reviewers seems to be the lack of comparisons with the strongest baselines in adversarial robustness, which as the authors argue mostly exploit additional data to further boost robustness. In this sense, I symphatize with the intuitions of the authors that DyART is improving robustness in an orthogonal way, and will probably further improve robustness in those settings. However, this is still a hypothesis, and should be tested to fully confirm this. In this regard, it is important to mention that these stronger baselines are not necessarily much heavier in terms of compute (roughly 4 times more than the AT setting in this work) so it would have been feasible to provide some experiments with them in this work.
> > >
> > > Thanks for pointing out the run time for methods using additional data! We ran the setting with additional data using the code base (for paper [1]) here : https://github.com/imrahulr/adversarial_robustness_pytorch. We replace the Batch Normalization (BN) layer with Group Normalization (GN) in WRN-28-10 and run for 200 epochs. For DyART we directly apply the same hyperparameters as in our paper and for TRADES, we apply the hyperparameters suggested by [1]. The results are as follows:
> > >
> > >
> > > | Defense | Clean | $\epsilon=4/255$ | $\epsilon=8/255$ | $\epsilon=12/255$ | $\epsilon=16/255$ |
> > > |:-------- |:--------:| --------:|--------:|--------:|--------:|
> > > | TRADES     |  84.93   |     72.41 |    56.07 |    38.15 | 25.47 |
> > > | Ours     |   86.57   |     73.38 |    56.85 |    39.2 | 25.92 |
> > >
> > > We can see that when utilizing additional data (1M sythetic data from a diffusion model), the performance of both methods improves and DyART still performs better. Due to the limited time, we do not tune our hyperparameters for DyART in this setting, and the performance results can be potentially improved by experimenting with different hyperpameters.
> > >
> > > We are aware that the number shown above is not the highest in the benchmark, due to the usage of GN instead of BN: we experiment with TRADES using BN and same hyperpameters and the robust accuracy is 59.38% under $\epsilon=8/255$, which is better than the GN case. However, we would like to stress that when using the same architectures with GN, DyART still performs better than TRADES in the setting of using additional data. We will definitely work on adapting our method to BN in the future.
> > >
> > > ---
> > > Reference:
> > >
> > > [1] Gowal, Sven, et al. "Uncovering the limits of adversarial training against norm-bounded adversarial examples." arXiv preprint arXiv:2010.03593 (2020).

---

> > > > ### Comment · Reviewer_1WcD · 2022-11-18
> > > > **Thank you for your reply**
> > > >
> > > > Thank you very much for taking the time to run these baselines. The results do look very promising indeed!
> > > >
> > > > After seeing these results, I have no remaining serious issue with this paper. Personally, I think this work is a strong contribution to the community, and I will increase my score to highlight my opinion of it.

---

> ### Author Response · Authors · 2022-11-14
> **Adding ablation experiments and relevant literature**
>
> We thank Reviewer 1WcD for the detailed and insightful feedback. We are inspired and encouraged that Reviewer 1WcD finds our ideas fresh and effective, our mathematical derivations brilliant, our writing clean and our experimental results competitive. Below we address Reviewer 1WcD's concerns in detail.
>
> ---
>
> >  Q1: Lack of ablations on new method: I find the paper would be stronger if, besides reporting best performances on two datasets, it also described how different hyperparameters affect the performance and dynamics of their proposed method. How do hyperparameters affect performance? And the burn-in period? ... It would also be nice to provide runtime numbers of DyART and compare them to the competing methods.
>
> Thanks for the suggestion! We have included the hyper-parameter sensitivity experiments in Appendix D.2. Here we briefly summarize the ablation studies:
> recall that the cost function for robustness is of the form $h(R) = \frac{1}{\alpha}\exp(-\alpha R)$ when $R < r_0$ and $h(R) = 0$ otherwise.
>
> 1. $\alpha$ and $r_0$ collectively determines the cost function: Larger $\alpha$ and a smaller $r_0$ corresponds to a cost function $h(\cdot)$ that decays faster, and we found that they will result in higher robust accuracy under smaller $\epsilon$ and also higher clean accuracy, while the robust accuracy under larger $\epsilon$ will be lower.
>
> 2. The robust loss coefficient $\lambda$: larger $\lambda$ results in higher robust accuracy.
>
> 3. The burn-in period: we suggest using a relatively large learning rate for natural training for the burn-in period (e.g. 0.1 for CIFAR-10 and Tiny-ImageNet).
>
> As for the run time number, we use the torch.cuda.Event functionality in PyTorch to measure the execution time for one iteration of each method. We use ResNet-18 with GroupNorm on a batch size of 128 on the CIFAR10 dataset. We use one NVDIA RTX A4000. The results are as follows:
> 1. natural training: $46 \pm 0.9$ ms
> 2. AT (PGD-10 on Cross-Entropy loss): $531.4 \pm 5.2$ ms
> 3. TRADES (PGD-10 on KL divergence loss): $573 \pm 2.8$ ms
> 4. DyART (10 steps for finding the closest boundary point): $743 \pm 10.3$ ms
> 5. DyART (20 steps for finding the closest boundary point): $1171 \pm 17.8$ ms
>
> Finding the closest boundary point (i.e. solving a constrained optimization problem) is important for DyART and developing more efficient method is one of our future directions.
>
> ---
>
> > Q2: (Minor) Missing relevant citations: In general, I find the paper is missing citations to a few relevant works that also have studied the dynamics of the margins of adversarial training or been inspired by them to improve the performance of robust models
>
> Thank you for pointing out these relevant works! We have updated the related work section (section 2) and also provide more related work in Appendix A.

---

### Official Review · Reviewer_rwTE · 2022-10-24

**Confidence:** 3
**Correctness:** 3
**Technical Novelty And Significance:** 2
**Empirical Novelty And Significance:** 2
**Recommendation:** 8

**Clarity, Quality, Novelty And Reproducibility:**


### Clarity
I have no problem regarding the clarity of the paper. It is well-written.


### Quality
The theoretical parts are sound; however, the empirical part can be improved to be stronger.


### Novelty
Firstly, I think the observation of the sign of boundary speed and the weighting function $h$ to balance large and small margins are novel contributions. However, the idea to  design an algorithm that pushes boundaries away is in fact pretty common. As a result, the paper may have not given enough credits to the existing work and form a detailed discussion. For example, the robust loss in TRADES is in fact a way to push boundaries away and the KL-divergence between clean and adversarial logits is roughly a way to penalize more on points that are closer to the boundary. I think this can be an explanation why TRADES results are so close to the new method in Table 1. Can the authors also provide the boundary speed comparison between TRADES and DyART? I wonder how much better DyART improves on pushing nearby boundaries away compared to TRADES. Moreover, this work [1] directly pushes the nearby hard boundaries away. Although the focus of [1] is certifiable robustness instead of the empirical one, it should at least be discussed in the paper.


[1] Croce, Francesco et al. “Provable Robustness of ReLU networks via Maximization of Linear Regions.” AISTATS (2019).



### Responsibility
A statement on where to find ways to reproduce the experiment can better improve the reproducibility of the work.


**Strength And Weaknesses:**

### Strength

**Presentation.** The presentation of the paper is very clear and easy to follow. Thank you for the writing.

**Motivation.** This paper is well-motivated. The problem of pushing decision boundaries away from points that are hard to produce robust predictions has existed for a long time. The proposed solution is very intuitive and the key message is easy to grasp (thanks to the clear writing).

**Theoretical Contributions.** Theorems characterizing the boundary speed and soft boundaries are important contributions. These are very interesting results and measurements, potentially can be used by future researchers for similar topics.

### Weakness
My biggest concern of this work is the empirical evaluation of the paper. I also have some minor concerns regarding the novelty of the work (I will elaborate this in the next review box).

Firstly, my biggest concern on the empirical results is simply that the baseline robust accuracy on CIFAR-10 (i.e. AT) is too low for a new method proposed in 2022. Although the improvement of the robust accuracy compared against the baseline numbers still show the new method is effective, they are still outside the ballpark of the state-of-the-art numbers. For example, on the benchmark page (https://robustbench.github.io/), the top-15 entries on CIFAR-10(linf, 8/255) have robust accuracy higher than 60% while all methods reported in this paper are below 50%. Even though I don’t think a paper has to outperform the state-of-the-art results to get accepted (and personally I am very enjoying reading and studying the solution proposed in this paper), empirical results at least within the ballpark will make the experiments stronger. I think the same problem happens to the Tiny-ImageNet results.

Secondly, there are a few hyper-parameters in the proposed method. It would be nice to have some sensitivity experiments to tell how the results are affected by these parameters and what are the recommended numbers for any follow-up work to use.


**Summary Of The Paper:**

This work presents a new robust training approach by pushing the nearby soft boundaries away for points that are closer to the boundaries. The motivation of the paper is an observation that some points are not pushed away from the boundary in the existing PGD training; therefore, it is straightforward to design an algorithm that is aware of the decision boundary. The authors test their method on CIFAR-10 and Tiny-ImageNet to show the effectiveness of the work.


**Summary Of The Review:**



In summary, I am currently on the fence. I like the theoretical part of the paper but the empirical part is not that convincing to me. I will vote for a weak accept for now but may be happy to raise if my concerns are resolved.

---

> ### Author Response · Authors · 2022-11-14
> **Our novelty is to directly implement the intuition of pushing the decision boundary away**
>
>
> > Q3.1 However, the idea to design an algorithm that pushes boundaries away is in fact pretty common. As a result, the paper may have not given enough credits to the existing work and form a detailed discussion. For example, the robust loss in TRADES is in fact a way to push boundaries away and the KL-divergence between clean and adversarial logits is roughly a way to penalize more on points that are closer to the boundary. I think this can be an explanation why TRADES results are so close to the new method in Table 1. Can the authors also provide the boundary speed comparison between TRADES and DyART? I wonder how much better DyART improves on pushing nearby boundaries away compared to TRADES.
>
> Since the goal of $l_p$ robust training is to maximize the proportion of margins that are larger than the attack length, almost every robust training method (including TRADES) pushes decision boundary away. However, previous methods push away the decision boundary based on heuristic. For example, the KL-divergence (computed by the **output** logits of the neural network) is a surrogate of the margin in the **input** space, though it is very likely that smaller KL-divergence does not correspond to smaller margin.
>
> However, in this work, we are curious about the intriguing question: Is it possible to **directly** push the decision boundary away, and directly maximize the margin defined in the **input** space? Our work shows that this is in fact possible by providing a framework for decision boundary dynamics and extensive experiments.
>
> In the paper, we do not claim that we are the first method that pushes the decision boundary away. Our contribution is to **directly** manipulate the input margins, providing a more explicit solution to robust training.
>
> As for the comparison between DyART, TRADES and AT, here we provide the proportion of negative speed among points with margins smaller than $\frac{4}{255}$ on the naturally pretrained model: 15.3% (DyART) < 22.8% (TRADES) < 29.2% (AT). Note that the experimental setting is the same as in Section 6.2, only that we additionally collect the dynamics for TRADES (with $\lambda=6$). Since the robustness performance is DyART>TRADES>AT, We conclude that less conflicting dynamics does help training robust models.
>
> ---
>
> > Q3.2 Moreover, this work [1] directly pushes the nearby hard boundaries away. Although the focus of [1] is certifiable robustness instead of the empirical one, it should at least be discussed in the paper.
>
> Thanks for mentioning the certified robustness literature! Since in this paper we consider the exact margin, our literature review mainly focuses on maximizing the margin. We added a dedicated paragraph on certified robustness method in the Appendix A to further explain the relationship between our work. In a word, certified robustness radius is a (possibly loose) lower bound of the margin, and our work is concerned with the margin itself.
>
> ---
>
> Thank you again for your time and effort in reviewing our paper! Please let us know if the above explanations do not address your concerns. We are happy to answer any further questions.

---

> ### Author Response · Authors · 2022-11-14
> **Clarification on empirical results and sensitivity experiments**
>
> We thank Reviewer rwTE for the detailed and insightful feedback. We are encouraged that Reviewer rwTE finds our framework interesting and useful for future research, and considers our paper well-motivated as well as well-written. Below we address Reviewer rwTE's concerns in detail, and provide additional experimental results.
>
> ---
>
> >  Q1: Firstly, my biggest concern on the empirical results is simply that the baseline robust accuracy on CIFAR-10 (i.e. AT) is too low for a new method proposed in 2022. Although the improvement of the robust accuracy compared against the baseline numbers still show the new method is effective, they are still outside the ballpark of the state-of-the-art numbers. For example, on the benchmark page (https://robustbench.github.io/), the top-15 entries on CIFAR-10(linf, 8/255) have robust accuracy higher than 60% while all methods reported in this paper are below 50%. Even though I don’t think a paper has to outperform the state-of-the-art results to get accepted (and personally I am very enjoying reading and studying the solution proposed in this paper), empirical results at least within the ballpark will make the experiments stronger. I think the same problem happens to the Tiny-ImageNet results.
>
> On the benchmark page, the most significant factor for boosting the robustness performance is the usage of more data (either additionally collected data or synthetic data from generative models[1]). While this line of works focuses on how to utilize more data (and they still use TRADES[2] as the base robust training method) and achieves impressive improvement, our work is orthogonal to them: we focuses on designing a robust training method that directly increase margins and prioritize increasing smaller ones.
>
> We added this line of work and compare them with our method in Appendix A.
>
> ref:
> [1] Gowal, Sven, et al. "Improving robustness using generated data." Advances in Neural Information Processing Systems 34 (2021)
> [2]Rebuffi, Sylvestre-Alvise, et al. "Fixing data augmentation to improve adversarial robustness." arXiv preprint arXiv:2103.01946 (2021).
>
>
> ---
>
> > Q2: Secondly, there are a few hyper-parameters in the proposed method. It would be nice to have some sensitivity experiments to tell how the results are affected by these parameters and what are the recommended numbers for any follow-up work to use.
>
> Thanks for the suggestion! We have included the hyper-parameter sensitivity experiments in Appendix D.2. Here we briefly summarize the ablation studies:
> recall that the cost function for robustness is of the form $h(R) = \frac{1}{\alpha}\exp(-\alpha R)$ when $R < r_0$ and $h(R) = 0$ otherwise.
>
> 1. $\alpha$ and $r_0$ collectively determines the cost function: Larger $\alpha$ and a smaller $r_0$ corresponds to a cost function $h(\cdot)$ that decays faster, and we found that they will result in higher robust accuracy under smaller $\epsilon$ and also higher clean accuracy, while the robust accuracy under larger $\epsilon$ will be lower.
>
> 2. The robust loss coefficient $\lambda$: larger $\lambda$ results in higher robust accuracy.
>
> 3. The burn-in period: we suggest using a relatively large learning rate for natural training for the burn-in period (e.g. 0.1 for CIFAR-10 and Tiny-ImageNet).

---

> > ### Comment · Reviewer_rwTE · 2022-11-18
> > **Response to Authors**
> >
> > Thank you for your response. I have just checked out the new paper and it solves most of my concerns. I deeply undertstand that this work focuses on an orthogonal direction other than those data augmentation tricks used in boosting the adversarial robustness, especially on CIFAR-10. I personally also agree with the authors that DDPM data is the key to improve the robust accuracy to some like 60%. But, it would really be nice to have those DDPM data ready for all baselines and the new method in the main body of paper and show that your method can achieve better results on CIFAR-10. It is really easy for people who do not deeply understand the technique of the work to overlook its merit if the shown robust accuracy is much lower than the recent numbers. However, I don't want to overlook the value in this work so I will increase the scores but I sincerely suggest the authors to run experiments with those techniques that boost the robust accuracy on the baseline, e.g. PGD/TRADES training, to a reasonably good number. After that, you can apply your technique and show how much better it is.

---

> > > ### Author Response · Authors · 2022-11-19
> > > **Thank you for your comments**
> > >
> > > Thank you for your follow-up comments and for appreciating the value of our work! Indeed using additional data is important and we will definitely explore this direction for our proposed method in the future.

---

### Official Review · Reviewer_xLGt · 2022-10-31

**Confidence:** 3
**Correctness:** 4
**Technical Novelty And Significance:** 3
**Empirical Novelty And Significance:** Not applicable
**Recommendation:** 6

**Clarity, Quality, Novelty And Reproducibility:**

Clarity: Good.

Quality: Good.

Novelty: Ordinary.

Reproducibility: No comment.

**Strength And Weaknesses:**

Strength:
1. The motivation is interesting and heuristic, pointing out the remaining shortcomings of adversarial training.
2. The idea is simple and clear, for mitigating the concerning robustness issue.
3. The presented experimental result outperforms previous works (2021 or ealier) in similar area.

Weaknesses:
1. Almost every individual idea is too intuitive, so that none of them is groundbreaking or novel.
2. A few experimental results shown. Maybe try some large-scale multiclass dataset like ImageNet.
3. Some sentential expression is not very clear: at the end of 2nd/3rd paragraph of Section 2, the conclusions are likely the same.

**Summary Of The Paper:**

This paper provides another viewpoint of investigating the robustness of DNNs. The motivation is that when a model is training, the decision boundary w.r.t. the model is dynamically changing over epoch time, so that the distance of training data points and the decision boundary varies. When the distance gets closer, that data point becomes vulnerable. Due to this motivation, the paper defines some functions such as “curve of the closest boundary point” and “speed of the decision boundary”, that are helpful to visualize the phenomenon of the dynamics of decision boundaries, and to alert which data point may not be safe. Moreover, this paper proposes a robust training algorithm, called Dynamics-Aware Robust Training (DyART), to encourage the decision boundary leaving away from vulnerable training data points. By combining the close-form for the gradient of smooth cost function of the margin, and the soft decision boundary, into the loss function, DyART can be trained efficiently, and can take both clean and robust accuracies into account.

**Summary Of The Review:**

Overall, I think that this is a good paper. It starts from the nature of dynamics of decision boundary, pointing out some shortcomings of existing robust model. Then, it proposes a training algorithm, called DyART, to mitigate such issue. With some experimental results, DyART is verified to be an effective robust model.

---

> ### Author Response · Authors · 2022-11-14
> **The novelty of our work is implementing the intuition into a practical and useful framework**
>
> We thank Reviewer xlGt for the detailed and insightful feedback. We are encouraged that Reviewer xlGt finds our proposed method simple, interesting and effective. Below we address Reviewer xlGt's concerns in detail.
>
> ---
>
> >  Q1: Almost every individual idea is too intuitive, so that none of them is groundbreaking or novel.
>
> One of our major contributions is to cleverly execute the intuition into a practical framework (closed-from expression for decision boundary dynamics) and a robust training method (directly increasing the margin by following its gradient using our closed-form expression). We believe that this is actually a strong **advantage**, since our method starts from a reasonable intuition that was technically difficult to deal with in previous work, and then provides an effective solution. In the following, we elaborate on the novelty of our paper.
>
> The intuitive idea is that, since ultimately robust training methods push the  decision boundary away and increase the margins, studying the speed of the decision boundary is important. However, **it was previously unclear how to compute the speed and increase the margins directly**. Therefore, previous methods were not able to explicitly execute such intuition. Instead, they adopt other approaches such as a min-max framework: e.g. minimize the maximal loss around data points. However, it is not necessarily true that larger loss around data point means the decision boundary is closer, since the loss is computed using the output of the neural networks, and the margin is measured in the input space.
>
> On the other hand, our proposed framework directly track and manipulate the decision boundary dynamics by **mathematically deriving tractable expressions**. Therefore, our method has really strong intuitions and also different from previous methods and thus should be considered as novel.
>
>
> ---
>
> > Q2: A few experimental results shown. Maybe try some large-scale multiclass dataset like ImageNet.
>
> In this paper, we use two datasets: CIFAR-10 and Tiny-ImageNet. CIFAR-10 has fewer classes (10 classes) and Tiny-ImageNet has more classes (200 classes). Note that Tiny-ImageNet is part of the ImageNet dataset, which fits in our computational resource budget. Our proposed method achieves better performance than baselines on both CIFAR-10 and Tiny-ImageNet, which is sufficient to demonstrate its effectiveness. Many prior work on robust training use TinyImageNet instead of ImageNet, due to the computation overhead of adversarial training. TinyImageNet is already of larger scale compared with CIFAR-10 and has a lot more clases, and thus can demonstrate the effectiveness of our method.
>
> ---
>
> > Q3 Some sentential expression is not very clear: at the end of 2nd/3rd paragraph of Section 2, the conclusions are likely the same.
>
> Thank you for pointing out! We have revised these two paragraphs in Section 2.
>
> ---
>
> Thank you again for your time and effort in reviewing our paper! Please let us know if the above explanations do not address your concerns. We are happy to answer any further questions.

---

> ### Author Response · Authors · 2022-11-28
> **Does our response address your concerns?**
>
> Dear reviewer xLGt,
>
> As the first stage of the review discussion has ended, we would like to kindly ask you to review our revised paper as well as our response and consider making adjustments to the scores. Please let us know if there are any other questions. We would appreciate the opportunity to engage further if needed.
>
> Best regards,
>
> Paper3155 Authors

---

### Author Response · Authors · 2022-11-14
**Thank you for the reviews**

We thank all reviewers for their insightful questions and valuable feedback. We are encouraged that reviewers finds our ideas fresh (1WcD), our mathematical derivations sound and elegant (095F,1WcD,rwTE), our paper well-written (wA1W,rwTE,1WcD,xLGt) and our experimental results competitive (xLGt,1WcD).


We have addressed individual questions of reviewers in separate responses. In the revised version, we incorporated all reviewers' suggestions by adding more motivating examples, more comparisons to prior works, more experimental results and baselines, as well as more implementation details. Here we briefly outline the updates to the revised submission for the reference of reviewers.

### Paper Updates:

- **[Section2: Related Work]** We added more related work and fix some sentential expression issues. (wA1w, o95F, rwTE, xlGt, 1WcD)

- **[Appedix A - Additional Related Work]** We added more related work on several aspects of robust training including (1) Decision boundary analysis; (2) Usage of additional data and data augmentation; (3) Certifiable robustness. (wA1w, o95F, rwTE, xlGt, 1WcD)

- **[Appendix D.2 - Experiments]** We added more ablation study on the hyperparameter sensitivity analysis. (rwTE, 1WcD)

- **[Appendix D.3 - Experiments]** We added the analysis of decision boundary dynamics throughout the whole training process. (wA1W, o95F, 1WcD)

The updates of the paper are in blue.

---
We again greatly appreciate all reviewers' suggestions. We hope that our paper updates and responses have addressed reviewers' questions and concerns. Please let us know if there are further questions.

---

### Decision · Program_Chairs · 2023-01-20

**Decision:**

Accept: poster

**Justification For Why Not Higher Score:**

Although this paper presents a simple but interesting idea in adversarial robustness, the adversarial accuracy is not significantly higher than those of SOTAs.

**Justification For Why Not Lower Score:**

This paper contributes to study the dynamics of decision bounrady to realize adversarial robustness.
Basically, all reviewers agree that this paper is well-motivated with somewhat intuitive/simple ideas, all the concerns have been properly addressed, and all reviewers give positive comments and higher scores.

**Metareview: Summary, Strengths And Weaknesses:**

Inspired by the observation that the robustness of a deep neural network in the classification task can be characterized by its margins, i.e., the decision boundary's distances to natural data points and the fact that it is still unclear whether the existing robust training methods effectively increase the margin for each vulnerable point during training, the authors studied a continuous-time framework for quantifying the relative speed of the decision boundary with respect to each individual point.
Based on further visualizing the moving speed of the decision boundary under Adversarial Training, a moving-behavior is revealed in that the decision boundary moves away from some vulnerable points but simultaneously moves closer to others, decreasing their margins.
To alleviate these conflicting dynamics of the decision boundary, the authors proposed Dynamics-aware Robust Training (DyART) with preliminary results conducted on Cifar-10 and Tiny-ImageNet.
The authors claimed that the proposed method is orthogonal to the exosting works; robustness can boosted without using additional training data.

Basically, all reviewers agree that this paper is well-motivated with somewhat intuitive/simple ideas, all the concerns have been properly addressed, and all reviewers give positive comments and higher scores.

Since the reviewers reach a consistent consensus in appreciating the contributions of this paper, the AC makes a decision of ``accept''.




**Note From Pc:**

if the above contains the word "oral" or "spotlight" please see: "oral" presentation means -> notable-top-5% and "spotlight" means -> notable-top-25%. As stated in our emails, we are disassociating presentation type from AC recommendations